

# GRAIN – A Global Registry of Agricultural Irrigation Networks

Sarath Suresh[1], Faisal Hossain[1], Vimal Mishra[2], Nehan Hossain[3]

[1]Department of Civil and Environmental Engineering, University of Washington, Seattle, WA 98195, USA

[2]Civil Engineering, Indian Institute of Technology (IIT), Gandhinagar, India

[3] Bothell High School, Bothell, WA, USA

*Correspondence to*: Sarath Suresh (saraths@uw.edu)

**Abstract.**

Despite supporting roughly 40% of the world's agricultural output, irrigation canal networks remain a critical data gap in

global geospatial archives. Currently, there is no consistent geospatial database that documents the extent of surface-water irrigation canals. The GRAIN (Global Registry of Agricultural Irrigation Networks) dataset fills this gap by leveraging the potential of volunteered geographic information (VGI) from OpenStreetMap (OSM) and applying a machine learning based classification pipeline to distinguish canals from rivers and streams. A Random Forest classifier was trained on 20,000 samples of quality-controlled canal and river data using 5 engineered geometric and topographical features. The model achieved over

98% training accuracy, translating to a ~93.6% median recall on independent validation datasets for primary canals, with a mean positional offset of ~98m. The GRAIN dataset includes land cover maps and OSM tags to assign canal use cases, identifying over 3.8 million km of agricultural irrigation canals in 95 countries. There is marked regional concentration of agricultural canals with hotspots identified in Europe, South and Southeast Asia, and North America. Agricultural canal distribution also varied widely by climatic zones with over 65% in temperate and cold zones and approximately 22% in arid

regions. A canal density analysis normalized by cropland area highlighted smaller countries such as Finland, the Netherlands, New Zealand, and Egypt having the densest irrigation canals. While the global correlation between canal density and national cereal yields was found to be modest (r = 0.31), the GRAIN dataset suggests that the presence of well-developed surface-water infrastructure may positively influence agricultural productivity as seen in the Netherlands and New Zealand. GRAIN is now publicly available at https://doi.org/10.5281/zenodo.16786488 (Suresh and Hossain, 2025) under a CC-BY-4.0 licence.

Designed as a community-driven resource, GRAIN data bridges a long-standing gap, and opens new possibilities for evaluating irrigation efficiency, supporting climate adaptation, guiding infrastructure investments, and extending the value of new satellite remote sensing missions on surface water such as the Surface Water and Ocean Topography (SWOT).

**Keywords:** Irrigation Canals, Global dataset, Agricultural Water Management, OpenStreetMaps (OSM)





## 1. Introduction

Engineered irrigation canals have long served as the backbone of modern agriculture, especially in the arid and semi-arid regions of the world. Today, the scale of agricultural water use is staggering, with irrigation accounting for approximately 70% of total freshwater withdrawals (UN WWDR, 2024). Globally, more than 307 million hectares of cropland are currently equipped for irrigation, and over 40% of this area is served by surface water systems. Despite representing just 20% of total cultivated land, irrigated agriculture contributes to nearly 40% of the world's total food production (FAO AQUASTAT),
underscoring its disproportionate importance to global food security.

Throughout history, large-scale irrigation canal projects have dramatically transformed regional agriculture and economies, allowing large swaths of arid or semi-arid regions to sustain agriculture (Angelakis, 2020). The Nile Delta of Egypt, for example, has for millennia been transformed from desert into fertile farmland by an extensive network of canals distributing the Nile's flow (Westerman, 1919; Cookson-Hills, 2016). In India, the Indira Gandhi Canal project carries water from the
Indus Basin deep into Rajasthan's Thar Desert. It was built to "green" the desert and irrigate ~1.8 million hectares of formerly barren land, which now supports the cultivation of wheat, cotton, mustard and vegetables (Hussain and Mohammad, 2018). Another notable example is California's Central Valley Project (CVP) in the U.S., which was initiated in the 1930s. The CVP spans more than ~450 miles down the length of California's arid Central Valley region, delivering over 7 million acre-feet of water a year to irrigate roughly 3 million acres of farmland. It has turned California into the nation's richest agricultural region,
accounting for an estimated 25% of U.S.'s food output (Nash, 2000; Johnson and Cody 2015). Canal irrigation has also been one of the drivers of the Green Revolution, especially in mid-20th century Asia, supporting double cropping and the adoption of high-yielding variety (HYVs) of rice and wheat (Spielman and Pandya, 2010). HYVs required continuous and controlled irrigation inputs, which was challenging with the traditional rain-fed farming practices, leading to millions of hectares of land being equipped with canals and tube wells (Gupta et al., 2003). In India alone, by the late 1980's, canal systems were irrigating
over 39 million hectares of land, supporting two to three crop cycles per year in many regions (Prasad and Shivay, 2022).

Despite their notable importance, there is no comprehensive and global geospatial dataset on irrigation canals today. While a few countries maintain national-scale hydrographic inventories that include engineered canals, such as the National Hydrography Dataset (NHD) in the United States (U.S. Geological Survey, 2022), most nations lack publicly accessible data on the extent and configuration of their major irrigation networks. This represents a critical data gap, particularly in this era of
big data, where easy access to large scale Earth observation data has become the norm. Satellite missions such as the Surface Water and Ocean Topography (SWOT), Landsat series, Sentinel series, SMAP (Soil Moisture Active Passive), and the recently launched NISAR (NASA-ISRO Synthetic Aperture Radar), significantly expands our ability to monitor the Earth's hydrological and agricultural systems. These missions provide detailed information on surface water extent, elevation, soil moisture, evapotranspiration, vegetation changes, and land surface dynamics among other data. However, the absence of a
global irrigation canal dataset with consistent meta data and quality control constrains our ability to fully leverage these rich data streams for agricultural water management. For example, although SWOT systematically captures large rivers wider than





100 meters (Biancamaria et al., 2016), it lacks a dedicated product for irrigation canals, as many such canals fall below the observable width. Since irrigation canals serve as the physical link between surface water sources (e.g. natural lakes and reservoirs) and croplands, a globally consistent geospatial dataset can expand the value of earth observations from space by allowing better quantification and representation of the full water delivery chain for irrigated agriculture. Such a dataset, if made openly accessible and designed to support community-driven enhancement, not only strengthens the scientific foundation for water and agricultural systems research, but also fosters transboundary and cross-jurisdictional cooperation by encouraging data sharing and collaborative problem solving in water resource management (Sarfaraz et al., 2022).

## 1.1. Challenges for Global Irrigation Canal Mapping

Mapping irrigation canals using traditional remote sensing is inherently challenging. Many canals, particularly secondary branches and distributaries are essentially narrow spatial features, often only a few meters wide. Such features fall below the pixel sizes of moderate-resolution open-access satellites such as Sentinel-2 or Landsat, which have spatial resolutions of 10 to 30m. Even primary irrigation canals, which can be tens of meters wide, can be hard to distinguish from natural rivers in imagery. Seasonal variability adds further complexity, as many canals are filled with water only during specific irrigation periods and may remain dry or obscured for much of the year (Arancibia, 2025). Vegetation overgrowth along canal banks, including grasses, crops, or tree cover, can mask the water surface entirely, further complicating detection. In such cases, the spectral signature of canals can be confusing to distinguish from other features such as dirt roads and crop field boundaries. Terrain-based extraction methods, that primarily uses digital elevation models (DEMs) to analyse topographical variations and flow direction often fail to detect engineered irrigation canals (Liimatainen et al., 2015).

The most viable approach for extracting irrigation canal geospatial features involves the use of high-resolution satellite imagery or aerial photogrammetry, combined with manual delineation or automated detection using traditional machine learning or deep learning techniques. These methods have demonstrated strong potential in accurately identifying narrow, linear features such as canals, particularly when supported by sub-meter imagery or drone-acquired orthophotos (Zhao et al., 2025). However, access to such high-resolution data, whether from commercial satellites like Maxar's WorldView series, PlanetScope, and Skysat, or from unmanned aerial vehicles (UAVs), is often prohibitively expensive, especially when scaled regionally or globally. In addition to high licensing costs, the logistical complexity of data acquisition and the computational burden of large-scale image processing pose significant barriers to widespread adoption.

## 1.2. Leveraging VGI as a Scalable Alternative

The recent growth of Volunteered Geographic Information (VGI) now offers a promising avenue for scalable geospatial data collection. VGI refers to the collective generation and curation of geographic information by individuals and communities, with platforms such as OpenStreetMap (OSM) (OpenStreeMap contributors, 2024), WikiMapia, and Yandex Maps at the forefront of this movement. OSM, in particular, has become a rich, open-access repository for a wide array of geographic features, including roads, buildings, and hydrographic networks. In OSM, contributors digitize linear water features, such as



rivers, streams, canals, ditches, and drains, based on high-resolution satellite imagery, local knowledge, and auxiliary sources
such as government records. Edits are made using a standardized tagging schema and community-driven validation processes.
This decentralized, bottom-up approach has made OSM grow rapidly in coverage and detail. For instance, in a rural region of
Portugal, the total length of mapped streams and rivers in OSM increased by ~15% between 2014 and 2023, as volunteer
citizen scientists added smaller tributaries that were previously missing (Moneteiro et al., 2024).

Despite its growing utility, there are some notable limitations associated with using VGI datasets like OSM for irrigation canal
mapping. The most prominent concern is tagging inconsistency. Contributors, especially those without domain specific
knowledge often mislabel canals as rivers, streams, and drains, or vice versa. Furthermore, OSM also does not distinguish
between transport canals (waterways), such as the urban water routes in Venice or Amsterdam, and agricultural irrigation
canals. Additionally, the completeness and accuracy of mapped waterways can vary by region. Thus, although the utility of
VGI is immense, they require additional refinement through post processing, and intelligent classification to be reliably used
for large scale irrigation network generation.

This study introduces GRAIN (Global Registry of Agricultural Irrigation Network), a first of its kind open-access geospatial
dataset focussed specifically on global irrigation canals. GRAIN combines OSM's volunteered waterway data with a machine
learning based classification pipeline and land-use land-cover maps to overcome some of the limitations of VGI. Through this
hybrid approach, GRAIN provides an open-access, global-scale geospatial dataset of irrigation canals, serving as a critical
missing link in the water-agriculture data ecosystem. GRAIN can potentially support a wide range of emerging research in the
field of agricultural water management, such as satellite-data driven irrigation advisory systems (Khan et al., 2025), and remote
sensing-based evaluation of irrigation efficiency, and policy-focused assessments of water food trade-offs (Al Zayed and
Elagib, 2017). Such a dataset can also possibly improve the assessment of groundwater recharge from irrigation canals. By
improving spatial understanding of surface-water delivery networks, GRAIN can potentially enable more accurate assessments
of water distribution, irrigation performance, and agricultural resilience under a changing climate.

In the following sections, we present the development, validation, and global insights derived from the GRAIN dataset. Section
2 outlines the methodology used for dataset construction, including data sources, processing workflows, and dataset
structuring. Section 3 details the validation conducted across key canal networks worldwide. Section 4 highlights key statistics
and spatial patterns, offering a global overview of irrigation canal infrastructure. Finally, Section 5 discusses the broader
implications, current limitations, and future directions for the GRAIN dataset.

## 2. Dataset Development

### 2.1. Scope and Extent

Developing the GRAIN canal data for the entire globe is not optimal, as large portions of the Earth, such as Siberia, Northern
Canada, the Sahara Desert, Greenland, the Amazon rainforest, and central Australia, are unlikely to have irrigated agriculture

in the recent past or in the near future. As illustrated in Figure 1, global irrigation is highly concentrated in specific regions, with pronounced hotspots across South, Southeast, and East Asia, Europe, and parts of North America. To optimize processing resources, the study focused efforts on regions where irrigation canals are likely to exist and matter more for agricultural water management. FAO Irrigated Area v.5. data (Siebert et al, 2013) was first used to identify irrigation hotspots using its 'percentage of grid cell irrigated' band (red shaded regions in Figure 1). Then for every country, the percentage of total area

covered by irrigated cells is computed. This helped identify countries and regions that can be avoided in the GRAIN data processing workflow. For version 1.0., the GRAIN dataset targeted 95 countries with considerable irrigated land (shown in shaded green in Figure 1). Certain countries, such as Russia and Canada, were then further broken down into their sub regions to avoid large swaths of non-irrigated areas.

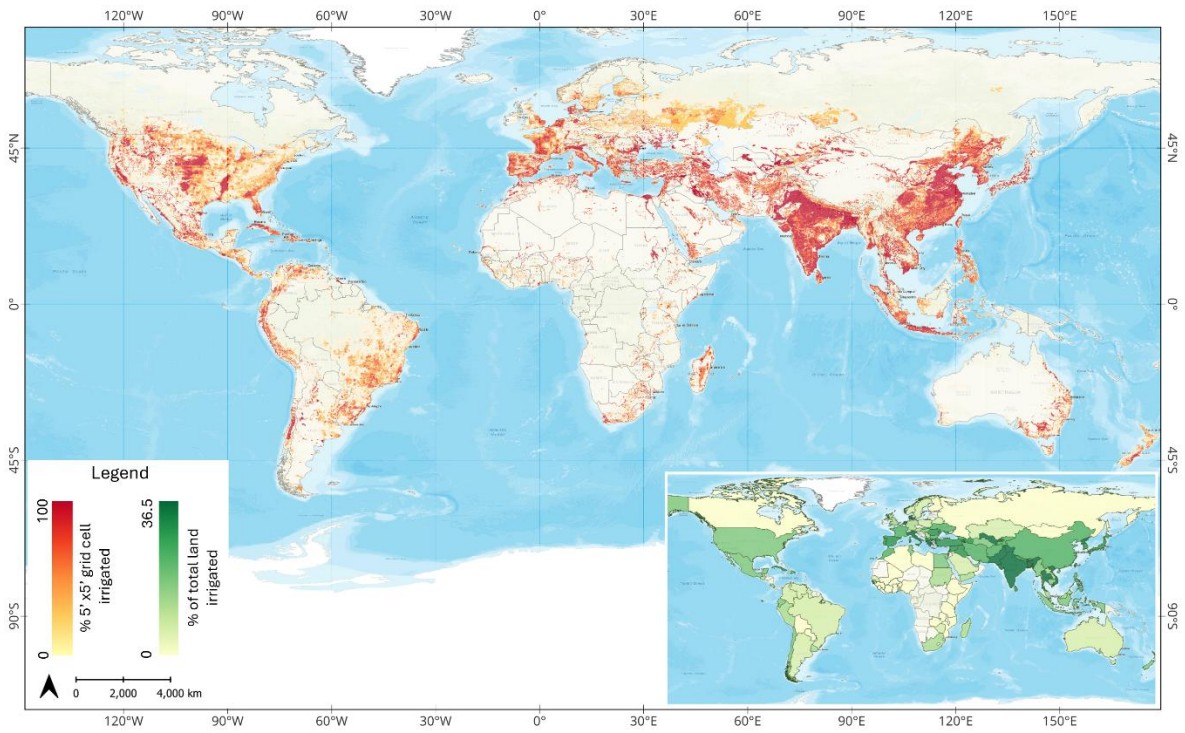

Figure 1. Global distribution of irrigated agriculture used to guide the spatial extent of the GRAIN dataset. Red shading indicates the percentage of each 5' × 5' grid cell under irrigation, while green shading seen in the inset panel represents the percentage of total national land area that is irrigated. Source: Irrigated area data - FAO Global Irrigation Area v.5.(Siebert et al, 2013); Country outlines - World Administrative Boundaries (World Food Programme, 2022); Basemap - Esri, DeLorme, HERE, USGS, Intermap, iPC, NRCAN, Esri Japan, METI, Esri China (Hong Kong), Esri (Thailand), MapmyIndia, Tomtom.





**2.2. Source Data**

The GRAIN dataset development relies on a combination of VGI data from OSM, satellite driven land cover products, hydrological river centreline products, and in-situ canal networks for the scalable identification of canal infrastructure. These datasets are utilised at various points in its workflow, as itemised in Table 1.

Table 1. Datasets used in the GRAIN workflow.

| Dataset Name | Type | Source | Purpose in Workflow |
|---|---|---|---|
| OpenStreetMap (OSM) | Volunteered GIS | OpenStreetMap contributors, 2025 | Primary source for hydrographic vector features. |
| FAO Global Irrigation Area v5 | Raster (5') | Food and Agriculture Organization of the United Nations (Siebert et al, 2013) | OSM data filtering for countries with significant irrigated land |
| ESA CCI Land Cover v2.0.7 (2015) | Raster (300 m) | ESA Climate Change Initiative (ESA, 2017) | Canal use case identification |
| SWORD v.15.(Surface Water and Ocean Topography River Database) | Vector (line and node points) | NASA JPL, University of North Carolina at Chapel Hill (Altenau et al., 2021) | Identifying natural river channels for training and post process filtering |
| In-situ Canal Network Data | Vector (line) | National datasets for U.S. (3DHP - NHD, U.S. Geological Survey, 2022), India Canal Dataset (Ministry of Jal Shakti, Department of Water Resources, 2022), Teesta Canal Project, Bangladesh Water Development Board (BWDB) | Training/validation of ML classifier |
| Manual Canal Delineations | Vector (line) | Created by authors | Validation of ML classifier |
| World Administrative Boundaries (ADM0) | Vector (polygon) | World Food Programme, 2022 ; OpenDataSoft | National boundary delineation for country-based processing |
| SRTM (Shuttle Radar Topography Mission) DEM | Raster (30m) | NASA (Farr et al., 2007) | Feature Engineering |
| HydroBasins v.1c. | Vector (Polygon) | HydroSheds (Lehner and Grill, 2013) | GRAIN Id creation and identification of SWORD reach. |
| Köppen- Geiger Climate Classification Map | Raster (5') | Climate Change & Infection Diseases, Vetmed Uni, Vienna (Beck et al., 2023) | Metadata |

OSM data is retrieved from the Geofabrik database (Geofabrik, GmbH, 2025), a structured distribution point for country scale OSM data, with daily updates, offering access to the latest OSM extracts without needing to download and process the massive terrabyte scale OSM planet file. Country scale OSM data is downloaded as OSM PBF files, an open-source binary format developed by the OSM community, for each of the 95 countries. OSM data for the U.S, and India are downloaded at the sub-



region scale due to the complexity and size of the country scale data causing issues with various tools in the OSM processing pipeline. Data for Canada, and Russia, are also downloaded as sub regional files, to avoid unnecessary processing over areas without irrigation. For Canada, data for British Columbia, Alberta, Saskatchewan, and Ontario regions are selected, while for Russia, data is retrieved for the North Caucasus, Central, and South Federal districts.

**2.2.1 In-situ and Manually Delineated Canal Data**

For training the machine learning based classification model and for subsequent validation, high quality reference datasets of irrigation canals are essential. GRAIN utilizes two national scale inventories, and other regional datasets including several manually delineated vector files for this purpose. The first national inventory used is the 3D Hydrography Program (3DHP) dataset, a component of the United States National Hydrography Dataset (NHD). This dataset provides detailed and curated vector representations of surface water features across the U.S., including canals, ditches, rivers, and natural streams, making

it a reliable source for model training and performance evaluation. The 3DHP was developed by integrating high-resolution elevation data from the U.S. 3D National Elevation Program (3DEP) with existing surface water features from the NHD, thereby enabling hydrography that is topographically aligned and hydrologically consistent.

The second national inventory used is the India Canal Network dataset (Ministry of Jal Shakti, 2022), acquired from the Open Government Data (OGD) Platform maintained by the Ministry of Jal Shakti and the National Water Informatics Centre

(NWIC). The canal network was delineated through on-screen digitization using merged satellite imagery from IRS LISS-IV and Cartosat, covering the period from 2008 to 2012. The dataset was further refined and validated using multiple ancillary sources, including 1:50,000-scale Survey of India (SOI) topographic sheets, SOI digital vector data, Google Earth Pro imagery, and Central Water Commission (CWC) index maps. This comprehensive approach makes the India Canal dataset a valuable training and validation resource for identifying irrigation infrastructure in diverse agro-hydrological settings.

For validation, canal data was also obtained from the Bangladesh Water Development Board (BWDB) for the Teesta Canal project. Additionally, few canal networks were manually delineated using QGIS, to provide further spatial coverage of validation datasets. These include well-known irrigated regions such as the Nile Delta in Egypt, the Lower Indus (Sindh)





region of Pakistan, and the Upper Indus (Punjab) region of Pakistan. Figure 2 illustrates the spatial distribution of these training and validation datasets, highlighting both national-scale inventories and regional canal networks.

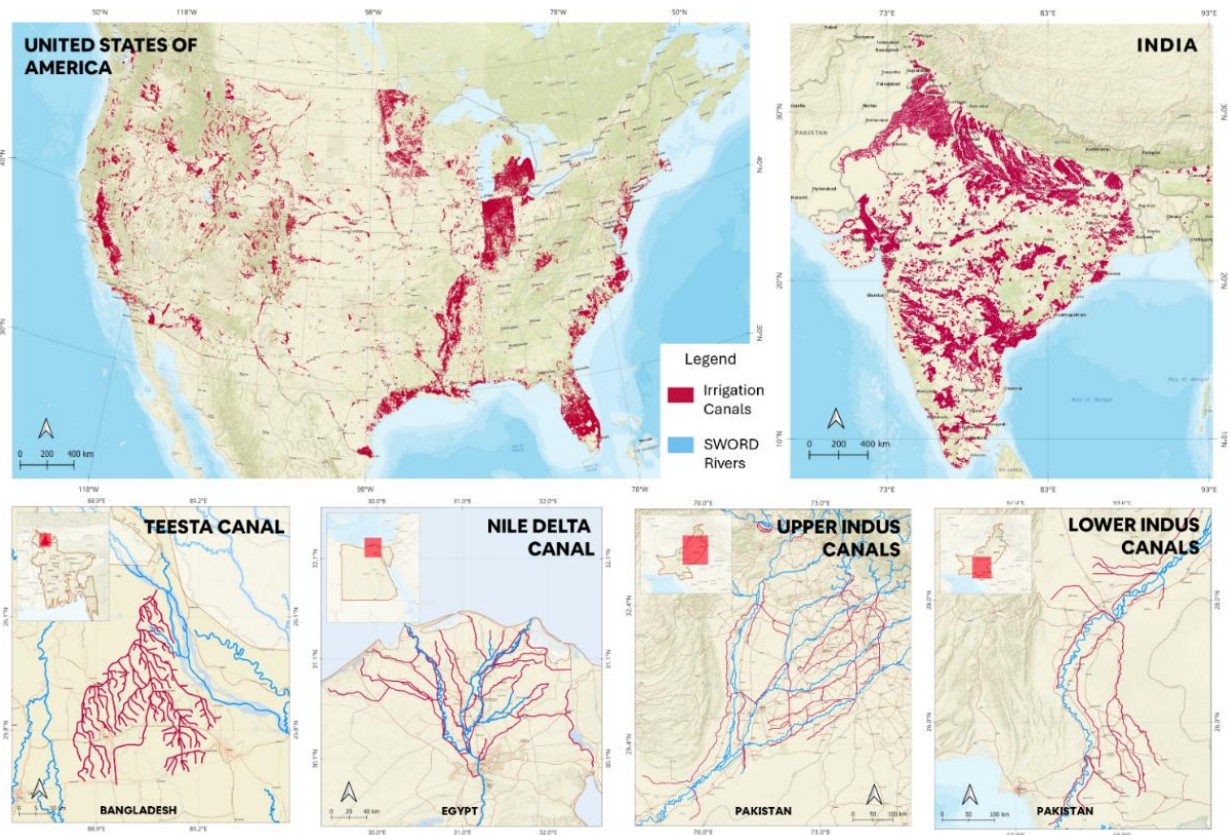


Figure 2. Training and Validation datasets used in the GRAIN development workflow. The top panels show national-scale irrigation canal inventories for the United States (left) and India (right). The bottom panels display regional canal networks including manually delineated canal networks used for validation: from left to right—Teesta River basin (Bangladesh), Nile Delta (Egypt), Upper Indus basin in Punjab (Pakistan), and Lower Indus basin in Sindh (Pakistan). Irrigation canals are shown in red, while SWORD river networks are shown in blue

for the regional scale datasets. Source: Canal dataset: U.S. – 3DHP NHD, India –Ministry of Jal shakti, Teesta – BWDB ; River centerline – SWORD v.15. ; Basemap - Esri, DeLorme, HERE, USGS, Intermap, iPC, NRCAN, Esri Japan, METI, Esri China (Hong Kong), Esri (Thailand), MapmyIndia, Tomtom.

### 2.3. Methodology

The end-to-end overview of the workflow used in the creation of the GRAIN dataset is shown in Figure 3. The process starts

with the extraction of the OSM data for countries having significant irrigated land, based on the FAO Global Irrigation Area v.5. dataset. Waterway features such as rivers, canals, stream, and drain, are then extracted from this country scale OSM data. These features then serve as input to a machine learning (ML) classifier trained to distinguish man-made irrigation canals from natural watercourses. The classification is supported by in-situ canal data, SWORD river centreline dataset (Altenau et al.,



2021), and land use/land cover (LULC) information from ESA's CCI product (ESA, 2017) to identify non-agricultural

channels. The output is a pre-validated canal dataset that undergoes statistical validation using both manually delineated canal maps and curated in-situ datasets from multiple regions. Finally, validated canal segments are assigned various metadata, to produce the final GRAIN dataset. This section details the various steps involved in each part of this workflow.

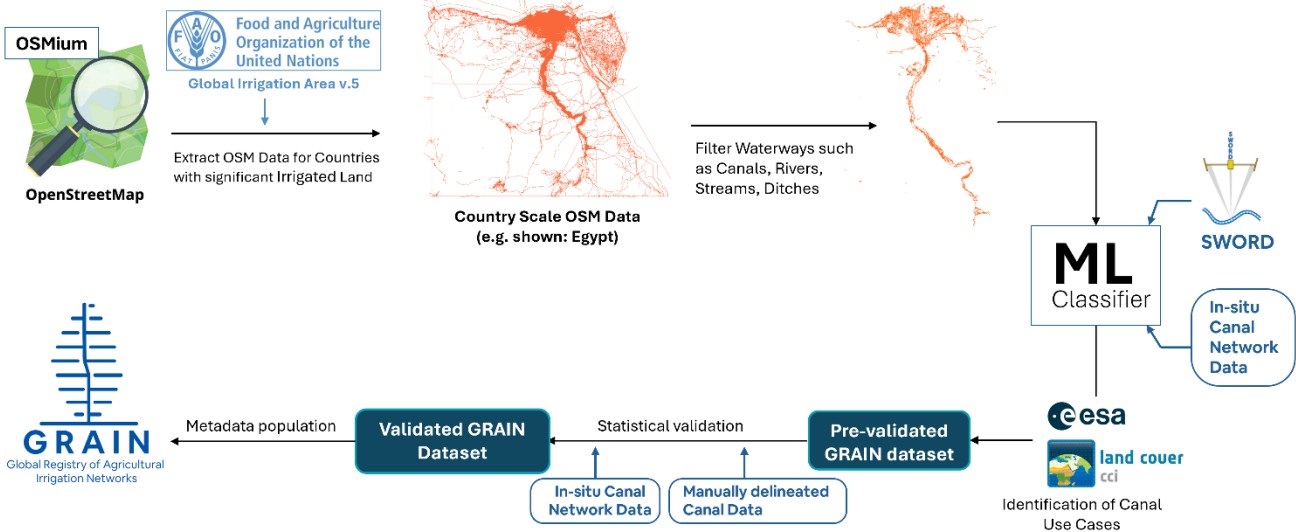

Figure 3. Workflow for the development of the GRAIN dataset. OSM data are extracted for countries with significant irrigated land (based on FAO Global Irrigation Area v.5), filtered for relevant hydrographic features, and classified using a machine learning pipeline informed by in-situ canal datasets, SWORD river networks, and ESA CCI land cover data. The resulting canal network undergoes validation and metadata population to produce the final GRAIN dataset.

### 2.3.1. OSM data retrieval and processing

The pipeline begins with the automated retrieval and processing of OSM data for the 95 countries covered by GRAIN. The python 'requests' package is used to programmatically access Geofabrik's download servers to retrieve country and regional scale '.pbf' data. These files are then processed using OSMium-tool, a command line interface (CLI) toolkit built on C++, to efficiently filter out only the 'waterways' line features. They are then fed into a QGIS based workflow to convert from '.pbf' to the lightweight GeoParquet format which can be easily digested by the subsequent python-based workflows.

Following conversion, each GeoParquet file underwent geometry cleaning to remove self-intersections and spurious ring closures. The geometries were then reprojected to a metric coordinate reference system (EPSG:3857) to facilitate distance-based computations for feature engineering. Finally, the cleaned dataset was split into two subsets, an OSM river dataset



containing features with the waterway tag of 'rivers', and 'streams', and OSM canal dataset containing those tagged as 'canal', 'drain', or 'ditch'.


### 2.3.2. Machine Learning Model Training

The objective of the machine learning (ML) component of the GRAIN workflow was to address the limitations of VGI, particularly the frequent mislabelling of canal and river features. To build a labelled training dataset, we used an intersection-based sampling strategy. Canal training samples were selected by identifying OSM canal features that spatially intersect with national canal inventories. River samples were derived from the OSM river dataset that intersect with known river centrelines in SWORD. This ensured that all training samples originated from OSM but were anchored to trusted reference datasets. A total of 10,000 canal samples and 10,000 river samples were then selected from these intersection datasets through random sampling, with care taken to ensure that the training samples did not spatially overlap with regions later used for validation.

The classification model used was a Random Forest, chosen for its balance between simplicity, accuracy, computational efficiency, and interpretability. Random Forests have proven to be well-suited for geospatial classification as their ensemble nature reduces overfitting risks (Breiman, 2001; Rodriguez-Galiano et al., 2012). They also have strong ability to handle high-dimensional data, and provides internal estimates of feature importance, which is valuable for interpretability in geospatial modelling applications (Belgiu and Drăguţ, 2016).

Feature engineering in the GRAIN workflow is performed using five carefully selected geometric and topographic properties designed to capture the structural differences between natural rivers and man-made canals:

- **Straightness ratio or Sinuosity**: The straightness ratio is defined as the ratio of the Euclidean distance between the start and endpoints of a polyline and its actual path length, with Sinuosity being the inverse of this property. It is bounded between 0 and 1 with values close to 1 indicating a straight line. The utility of straightness ratio for distinguishing irrigation canals, which tend to be more rectilinear, from natural rivers, has been demonstrated in small-scale studies, such as in the Mekong River Delta (Boon et al., 2025). In GRAIN, the conventional straightness ratio was adapted by computing the Euclidean distance as the sum of the straight-line distances between the start point (A) and midpoint (B), and the midpoint and endpoint (C), rather than the direct line from start to end. This better accounts for sharp deviations that engineered canals sometimes take. Figure 4 panel (a) and (b) showcases typical examples of an OSM river (blue line) and canal (red line). Note how the river is easily distinguishable to the human eye by virtue of its meandering geometry. If points A, B, and C are the start point, middle point, and end point of the polylines, then Straightness ratio is defined as:

$$Straightness\ Ratio = \frac{(D_{AB} + D_{BC})}{D_{path}} \qquad \text{Eq.1}$$





where $D_{AB}$, $D_{BC}$ are the Euclidean distances, and $D_{path}$ is the total path length of the polyline, measured along its geometry. From Figure 4, panel (c), OSM canals in the training data have significantly higher straightness ratio than rivers, owing to their generally straighter geometry.

- **Elevation Difference and Slope:** The 30 m SRTM Digital Elevation Model (DEM) was used to extract the elevation data at the start and end points of the polylines. The elevation difference is computed as the absolute difference between these two values. The slope is then calculated as the ratio of this elevation difference to the total path length of the polyline. Natural rivers in general exhibit much higher elevation differences than canals and a smaller but still significant difference in slope as seen in Figure 4 panels (d) and (e).

- **Mean Turning Angle:** It is defined as the average angular deviation between consecutive line segments in a polyline. It is computed by taking 3 vertices at a time along the polyline to form two vectors $\vec{u}$, and $\vec{v}$, and calculating the angle between them as:

$$\theta = arctan2(\ det(\vec{u}, \vec{v}), \vec{u}.\vec{v}) \qquad\qquad \text{Eq.2}$$

Where, $\vec{u}.\vec{v}$ represents the dot product, and $det(\vec{u}, \vec{v})$ is the 2D cross product. The mean turning angle is then computed as the average of all such angles along the polyline and is a measure of how much the polyline meanders. Natural rivers are expected to have a higher mean turning angle, as seen from the distribution plots in Figure 4, panel (f).

- **Curvature Index:** This feature quantifies the total angular change along a polyline relative to its length. It is computed as the sum of absolute turning angles (as described in Eq. 2), normalized by the total path length. In the GRAIN workflow, the curvature index is expressed as angular deviation per 100 meters of polyline length, as a means of standardising the feature. As seen in Figure 4 panel (g), higher values are indicative of more irregular or meandering paths, typical of natural rivers.



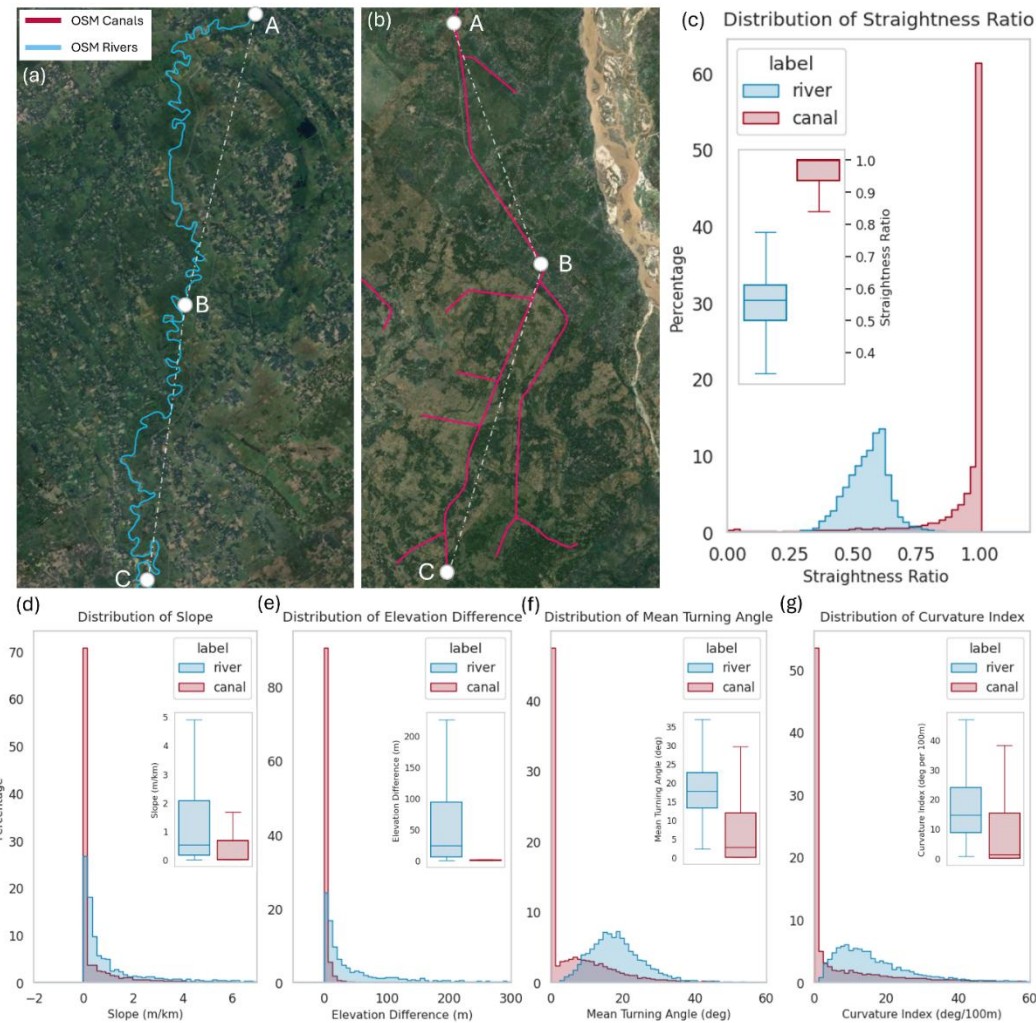

Figure 4. Geometric and Topographic properties used in the feature engineering of the Random Forest Classification model. (a) & (b)
Depictions of a typical OSM river (blue line) and canal (red line). Points A, B, and C represent the start, midpoint, and endpoint of the
vectors, and the dotted line indicates the straight-line distance between the points. Panels (c) – (g) presents histogram distributions for canal
and river samples in the training data, for 5 engineered features – Straightness ratio, Slope (m/km), Elevation difference (m), Mean Turning
Angle (°), and Curvature Index (°/100m). Insets within panels (c) – (g) shows the box plot distributions of the corresponding property.
Source: Canal and River geometry – OpenStreetMap (OpenStreeMap contributors, 2024); Basemap for panel (a) and (b) - Esri, DigitalGlobe,
GeoEye, i-cubed, USDA FSA, USGS, AEX, Getmapping, Aerogrid, IGN, IGP, swisstopo, and the GIS User Community.

The Random Forest classifier, configured with 200 decision trees, was initially trained and evaluated on the 20,000-sample
training dataset using two widely accepted techniques, an 80-20 train-test split followed by a 10-fold cross validation. Feature
importance analysis indicated that straightness ratio is the most significant feature in distinguishing between engineered canals
and natural rivers, followed by elevation difference and slope. Model evaluation metrics such as recall, precision, and F1-score





achieved values exceeding 98% across both validation schemes, indicating a strong classification performance and robustness

of the selected features. A detailed breakup of the classification performance testing, including diagnostic plots and confusion

matrices are provided in Appendix section A.1. Based on the performance results, the Random Forest classifier was retrained

on the full 20,000 sample dataset and exported as a '.pkl' file for subsequent deployment in the GRAIN workflow.

### 2.3.3. Classifying Country Scale OSM data

For each of the 95 countries, the trained Random Forest model is now applied on the cleaned OSM subsets, namely the OSM

canal dataset and the OSM river dataset. First, the five engineered features showcased in Sect.2.3.2., are computed for each

polyline vector in the datasets. These features were then passed through the previously trained Random Forest classifier to

predict class labels. Alongside the class label, the classifier also outputs a confidence probability, which was retained as a

measure of prediction certainty, and presented as an attribute field in the final dataset.

The OSM rivers dataset is then inspected for any polylines predicted as 'canals.' These polylines then undergo a series of

checks:

- SWORD overlap: If the polyline overlaps with any known SWORD river, it is assumed to be a natural river and is
  reclassified as such.
- Name and tag check: If the OSM 'tag', 'name', or 'name:en' field in 'alt_tags' contains the word 'river' but not
  'canal', the segment is also reclassified as a river.

- Intersection and confidence check: If the segment intersects a classified canal from the OSM canal dataset and has a
  model confidence score exceeding 0.8, it is promoted to a canal and added to the canal dataset.

A similar post-processing step was applied to the OSM canal dataset for segments classified as rivers. First, the name and tag

fields were reviewed. If they included the word 'canal' or its variants (e.g., 'canale' in Italian datasets), the segment is

reclassified as a canal. Remaining candidates were then subjected to a topological connectivity check, where their connectivity

to the end points of classified OSM canal segment were checked. If a segment classified as a river was directly connected to

the endpoint of one or more confidently labelled canals, it was classified as an irrigation canal. Since each newly promoted

segment may, in turn, connect to additional river-labelled segments, this topological propagation was repeated iteratively until

no further promotions occurred.

### 2.3.4. Canal Use Case detection and Metadata population

The OSM Canal dataset at this stage contained canals of multiple use types, such as agricultural, urban, and navigational

waterways, and those with other non-agricultural purposes. To assign canal use, the ESA Climate Change Initiative (CCI)

300m Land Cover product was utilized to check the majority land cover class around a given canal polyline. Each canal was

buffered by 1 km, and the dominant land cover class within this buffer zone is extracted. If the majority class corresponded to

cropland or irrigated agriculture, the canal was flagged as an irrigation canal. For canals not having a majority cropland class

in its buffered zone, an end-to-end connectivity check was done. If it connected to any irrigation canal, then it was assumed to



be part of the irrigation system and was classified as irrigation canals. If no connectivity was detected, then they were classified as urban canals, navigational waterways, or as others, depending on the major classes detected. Figure 5 visually illustrates this classification logic with two representative cases. The left panel shows Venice, Italy, where the dense canal network lies entirely within an urban land cover class and is correctly identified as non-irrigation. The right panel shows a segment of the

Suez Canal region in Egypt, where large portions of the canal traverse cropland and are correctly identified using the ESA CCI product.

At this point additional metadata is added to the GRAIN dataset. A unique GRAIN_id is created for every canal by combining the country iso3 code, Pfafstetter basin level 6 ID from HydroBasins, and sequential numbering. Other information such as

country, continent, Koppen-Geiger climate class code, source date, and version numbering is added.

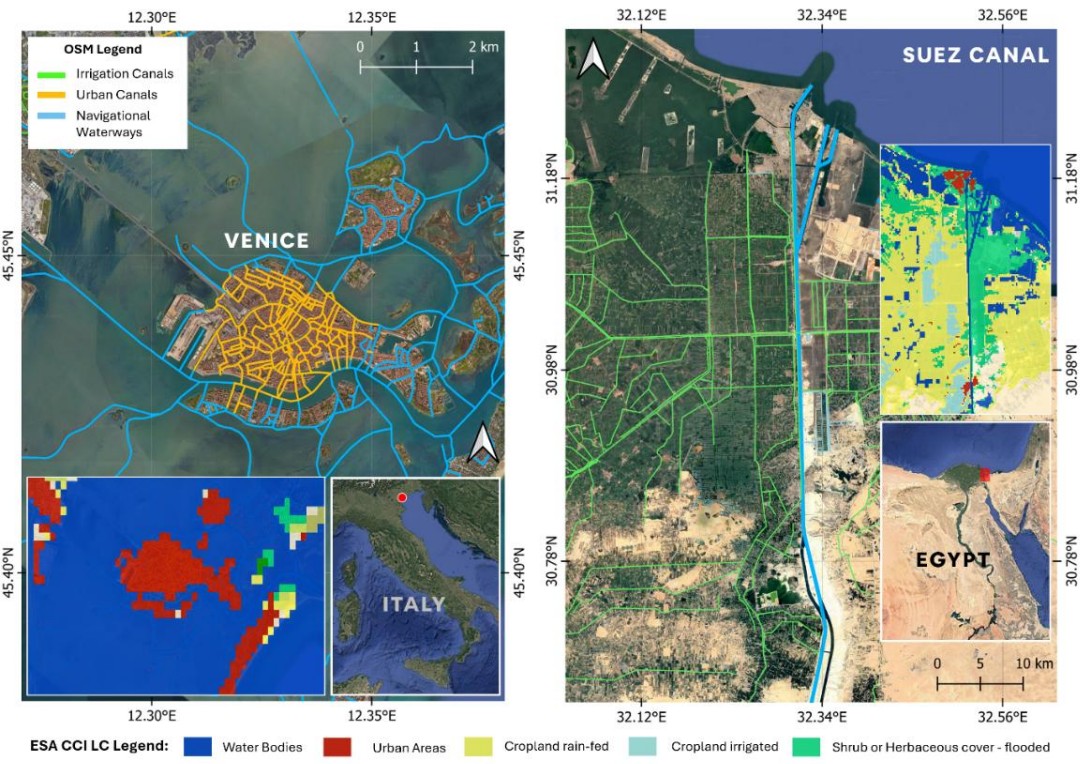

Figure 5. Determination of use case for OSM canals using ESA CCI Land Cover. (Left) The canal network of Venice, Italy. Yellow lines and blue lines are canals identified as being urban canals or navigational waterways. (Right) The Suez Canal (blue) and surrounding irrigation canals (green) in Egypt. The nearby irrigation network overlap regions dominated by cropland classes, highlighting their agricultural

function. Insets show ESA CCI land cover classes and regional context. Source: Basemap - Esri, DigitalGlobe, GeoEye, i-cubed, USDA FSA, USGS, AEX, Getmapping, Aerogrid, IGN, IGP, swisstopo, and the GIS User Community.



## 2.4. GRAIN Dataset Structuring

The GRAIN Dataset is distributed as country-scale files in two formats, a lightweight GeoParquet format and as ESRI Shapefiles, to ensure compatibility with legacy GIS software. All files are projected to EPSG:4326 based on the WGS-84 datum as per standard geospatial data convention. The attribute scheme for each GRAIN canal is given in Table 2.


Table 2. Attribute Schema for GRAIN canals

| Field | Type | Unit | Description |
|---|---|---|---|
| grain_id | String | - | Unique identifier – format: ISO3_PfafstetterL6ID_seq.numbering |
| osm_id | String | - | OpenStreetMap ID |
| country | String | - | The name of country where canal is located |
| continent | String | - | The continent where canal is located |
| country_iso | String | - | ISO-3 country code |
| length_KM | Float | km | Canal path length |
| slope_mkm | Float | $m\,km^{-1}$ | Longitudinal slope from SRTM DEM |
| elev_diff_M | Float | m | Elevation difference between start point and end point. |
| predicted_class | String | - | ML classified label ("canal", "river") |
| confidence | Float | - | Prediction certainty of ML classifier (0-1) |
| osm_label | String | - | Original OSM Label |
| osm_name | String | - | Canal name from OSM |
| alt_name | String | - | Any alternate canal name detected from OSM tags |
| tags | String | - | OSM tags if any |
| canal_use | String | - | Canal use case |
| koppen_class_code | String | - | Koppen-Geiger climate zone classification code |
| update_date | String | - | Date of dataset creation or update |
| version | String | - | GRAIN dataset release version number |

## 3. Validation

The validation of the GRAIN dataset was carried out to assess the geometric accuracy and completeness of the extracted irrigation canal networks. The performance of GRAIN canals was measured against in-situ and manually delineated canal vectors mentioned in Sect.2.2.1, using two performance metrics, Recall, and Mean Offset Distance (MOD) as shown in Figure 6. These metrics are reported separately for Primary and Secondary/Tertiary canals, based on the level of detail available in the reference datasets.


- Recall is a measure of how well the validation canals are represented in GRAIN. It is defined as the ratio of total length of validation canals that overlap with GRAIN canals to the total length of validation canals.






- MOD quantifies the positional deviation or the spatial offset between the validation canals captured by GRAIN and the GRAIN canals. The validation dataset was sampled at 100m intervals and the Euclidean distance to the nearest parallel GRAIN canal was considered as the offset distance. MOD was the average of all offset distances across the entire validation canal network, expressed in meters.

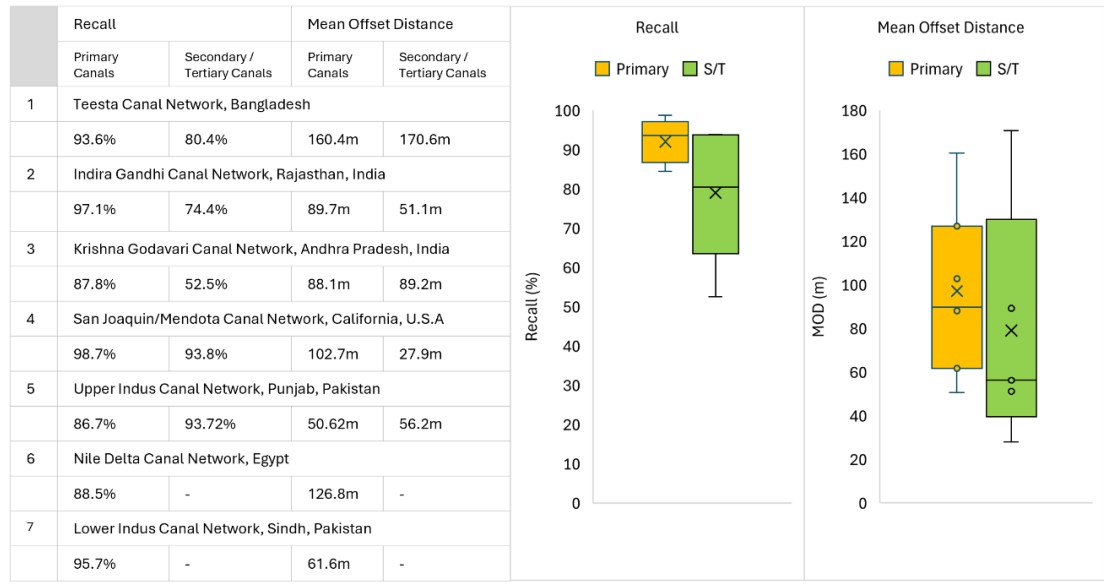


Figure 6. [left] Recall and Mean offset distance values for 7 validation canal networks, namely the Teesta canal in Bangladesh, the Indira Gandhi canal network, Rajathan, India, Krishna Godavari canal network, Andhra Pradesh, India, San Joaquin/Mendota canal network, California, U.S.A, Upper Indus canal network, Punjab, Pakistan, Lower Indus canal network, Sindh, Pakistan, and the Nile Delta canal network, Egypt. Values are segregated by Primary and Secondary/Tertiary (S/T) canals where available. [right] Box plots showcasing the

distribution of recall and MOD for Primary and S/T canals. Maps of each of the validation sites are provided in Appendix Figure A. 2.

Overall, GRAIN demonstrated good reliability in mapping large-scale primary canals, with an average recall of 92.6% and average MOD of 97.1m. In contrast, performance of secondary and tertiary (S/T) canals was more variable with recall ranging from 52.5% to 93.7%. S/T canals, however, was seen to have a lower MOD with an average value of 79m. These results highlight the usability of the GRAIN dataset for global or regional analysis, especially for major irrigation infrastructure, while

also indicating the need for cautious interpretation of S/T canal data in regions with sparse mapping and for granular (canal-level) decision-making.

## 4. Key Statistics and Global Patterns inferred from GRAIN

Globally, the GRAIN geospatial dataset identified a cumulative 3.82 million km of agricultural canals, the distribution of which is shown in Figure **7**. The United States has the highest total length of canals followed closely by countries such as

Germany, China, and India, which are well known for their extensive irrigation systems. A strong concentration is also



observed in eastern Europe, especially in Belarus, Poland, Russia, and Ukraine. Additionally, some countries across South and Southeast Asia, such as Indonesia and Pakistan also rank highly. This suggests a highly uneven distribution of the world's agricultural canals. GRAIN also identified ~63,000km of urban canals and ~39,000km of navigational waterways, with its distribution being presented in Appendix Figure A. **3**.

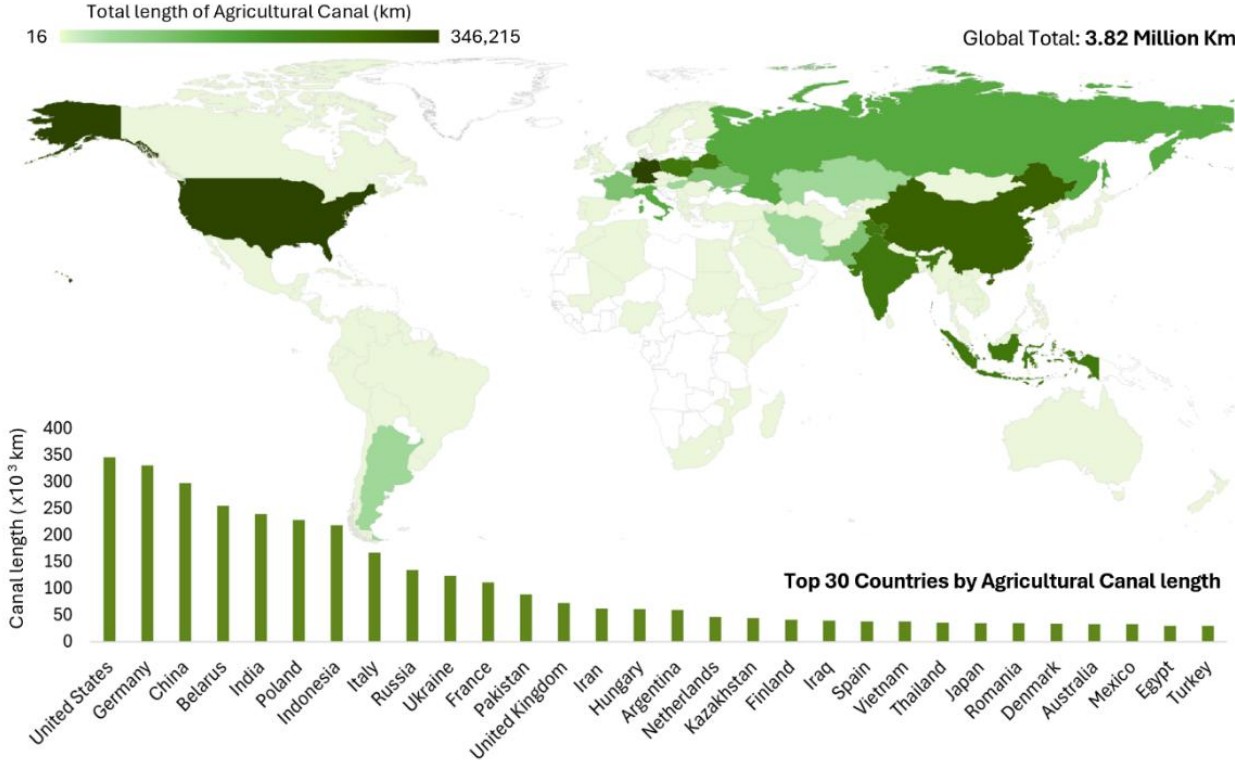


Figure 7. Global distribution of agricultural canals according to GRAIN dataset. [top] Choropleth map with countries having more extensive canal networks in a darker shade of green. [bottom] List of top 30 countries by total agricultural canal length. The U.S, Germany, China, Belarus, India, Poland, and Indonesia, together contain more than 60% of all canals.

The theme of uneven canal distribution continues when viewed through the lens of climate zones. Analysis using Koppen-Geiger climate zones (Beck et al., 2023), as shown in Figure 8, reveals that the majority of agricultural canals are located in cold and temperate climatic zones, accounting for 65.2% of all irrigation canals. In particular, the Dfb (cold, humid with warm summers), Cfb (temperate, humid with warm summers), and Cfa (temperate, humid with hot summers) climate classes dominate in terms of canal length. These zones correspond to regions across northern United States, Canada, Europe, and parts of China. Such regions have historically received significant investments in canal-based irrigation infrastructure to support agriculture during the cold winter months (Angelakis, 2020). Arid climatic zones such as BSk (arid cold steppe), BWh (arid hot desert), and BSh (arid cold desert), also account for considerable canal length, with a combined share of 21.9%. This highlights the important role played by irrigation canals in enabling productivity by compensating for rainfall deficits. In

contrast, the presence of canals in tropical zones is comparatively lower, at only 12.7%, with zones with monsoonal precipitation (Am) containing much lesser lengths of canals at just 1.3% of the global total. This suggests the possibility of a

prominently rain-fed irrigation system in such regions.

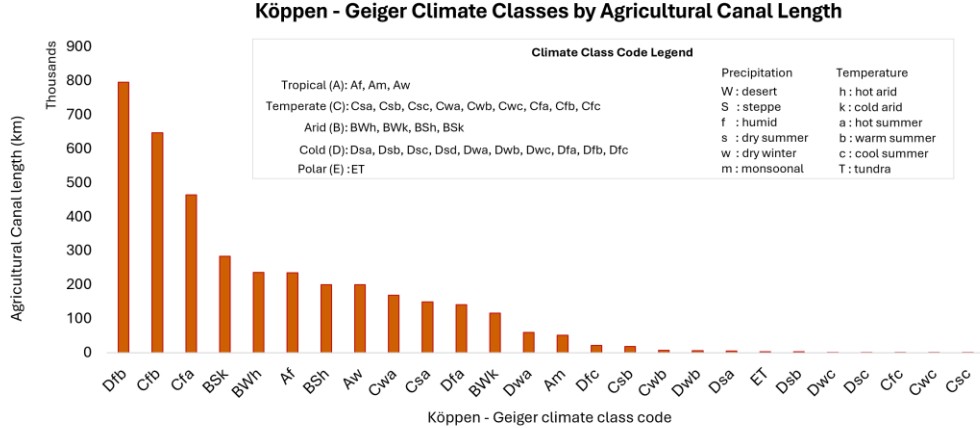

Figure 8. Distribution of agricultural irrigation canals in GRAIN by Koppen-Geiger Climate zones. Cold and Temperate zones such as Dfb (Cold, humid and warm summer), Cfb (Temperate, humid and warm summer), and Cfa (Temperate, humid and hot summer) contains the majority of canal networks. Readers are encouraged to refer to Beck et al., 2023 for details on these climate zone distributions.

A more comprehensive and nuanced pattern of irrigation infrastructure emerges from GRAIN when considering density relative to the total cropland available for irrigation. Figure 9 presents the variation of agricultural canal density globally, expressed as km of canal per 1000 km$^2$ of cropland, obtained by overlaying Global Food Security Support Analysis Data (GFSAD) Global crop mask dataset (Teluguntla et al., 2016) over GRAIN canals. As this metric normalizes canal distribution over cropland, it offers insights into how extensively irrigation canals are utilized for agricultural production. Despite having

modest overall canal length, countries such as Finland, Netherlands, Belarus, Taiwan, and New Zealand top the rankings, indicating highly developed and compact irrigation networks relative to their cropland extents. Extremely Arid regions, such as part of Africa, and Western Asia, with countries such as Egypt, Iraq, Iran, and the United Arab Emirates, also show very high canal density, showing their high reliance on irrigation canals.

Building on this perspective, Figure 10 used Cereal yield data (FAO, Ritchie et al., 2023), expressed as tonnes of produce per

hectare of cropland, to examine the relationship between agricultural canal density and food production. It can be observed that many countries with high irrigation canal density, such as the Netherlands, Germany, Taiwan, and New Zealand, also rank the highest for agricultural yield. While this relationship is not strictly linear, possibly due to the interplay of various other factors, a weak to moderate positive correlation can be observed (Pearson's correlation, r = 0.31). This trend highlights the potential link between well-developed irrigation infrastructure and improved crop productivity. Notably, several countries such

as the U.S.A, France, U.K, and China, exhibit very high yields despite moderate canal densities, indicating a strong presence



of alternate irrigation techniques and the adoption of efficient farming practices (Stubbs, 2016; Yan et al., 2020). Certain countries such as India and Belarus, showed lower yields despite having a large irrigation network or density, potentially reflecting regional disparities in agricultural water management. Overall, the analysis reinforced that while canal infrastructure is not the sole determinant of food productivity, its density remains an important indicator of agricultural yield potential.

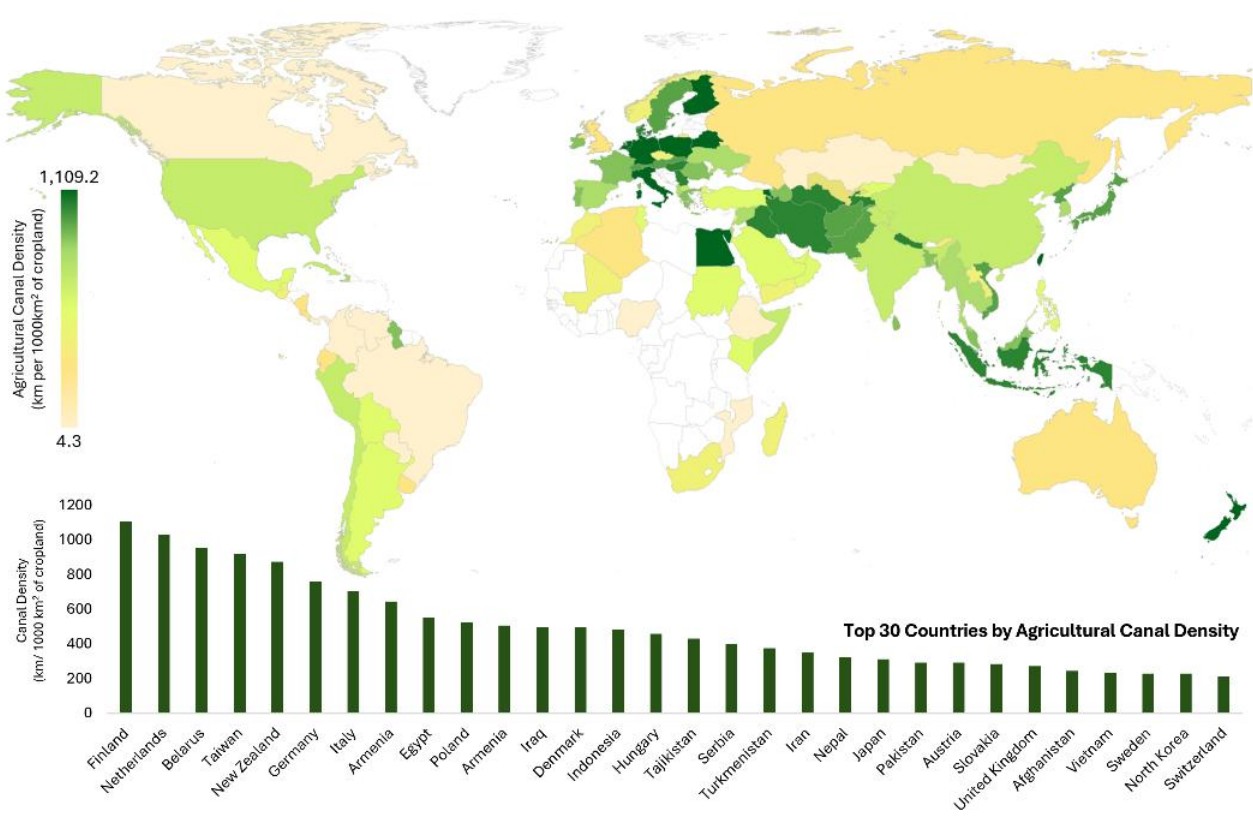


Figure 9. Agricultural Canal Density map constructed using GFSAD Crop Mask dataset (Teluguntla et al., 2016), expressed as km per 1000 km² of available cropland. [top] Choropleth map displaying the global variation of canal density. [bottom] Top 30 countries by canal density.

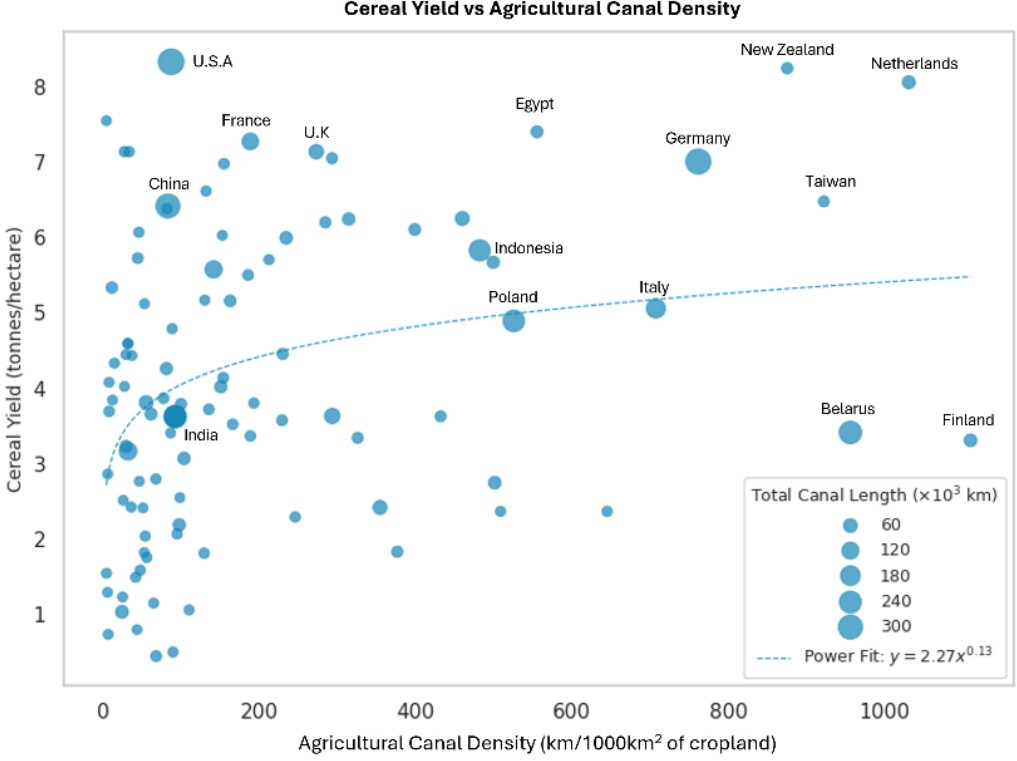

Figure 10. Agricultural Canal Density vs Cereal Yield as obtained from Food and Agricultural Organisation of the United Nations, processed
by Our World in Data (Ritchie et al., 2023). Cereals include wheat, rice, maize, barley, oats, rye, millet, sorghum, buckwheat, and mixed
grains. Data labels are provided for notable countries with high crop yield and canal density. The dotted blue line represents a logarithmic
best fit line indicating a positive trend. Marker sizes represent the total canal length.

## 5. Discussions and Conclusions

The critical role played by irrigation canals in sustaining agricultural productivity across diverse regions of the world is
unmistakable. The GRAIN dataset addresses a longstanding gap in the global water-agriculture data ecosystem by offering the
first open-source, globally consistent vector dataset of irrigation canal infrastructure, developed through a scalable and
reproducible workflow. By leveraging the rich and untapped data bank made possible via VGI, specifically OpenStreetMaps,
and refining it through a machine learning based classification pipeline, GRAIN overcomes many of the limitations associated
with traditional remote sensing approaches, which struggle to detect narrow, ephemeral, or vegetation-covered canals.

GRAIN's global irrigation statistics highlight striking patterns in canal infrastructure. While countries like the U.S., China,
and India dominate in absolute canal length, canal density metrics reveal more nuanced insights. Smaller nations such as
Netherlands, Taiwan, and New Zealand exhibit some of the most compact and intensely developed irrigation systems
sustaining exceptionally high agricultural yields. Similar cases are also seen in many arid and semi-arid countries such as





Egypt, Iraq, and parts of the Arabian Peninsula, displaying a highly concentrated and dense network of canal infrastructure,
despite relatively modest cropland areas. The temperate and cold climatic zone has emerged as a hot spot for irrigation canal
development, where extensive canal networks stabilize agricultural production during dry spells or winter shortages. Together,
these patterns highlight how both climatic necessity and agricultural intensification drive the spatial distribution of irrigation
canals worldwide.

The potential applications of this GRAIN dataset are many. For example, such a dataset can potentially help separate the role
of surface water and ground water in agriculture and help us understand the groundwater recharge from canal seepage. The
greening or browning trends observed in cropland today could potentially be explained by irrigation canal density and inform
policy-makers and planners on the further development of irrigation systems. The GRAIN dataset could also a key role in
creating satellite-based irrigation canal product on a global scale. Currently, the Surface Water and Ocean Topography
(SWOT) mission has demonstrated a capability to observe water conveyance systems as small as 50 m or less. Since SWOT
provides dynamic width, slope and also discharge as a point or reach-average value, it may be possible to have GRAIN dataset
embedded in SWOT data for the creation of a dedicate irrigation canal data with dynamic state of surface water in the canal
(discussed more below).

Validation of GRAIN dataset across several ground truth networks demonstrated reliably high spatial recall (>90%) and low
mean offset distances (<100m), confirming GRAIN's reliability for primary canals and for global or regional analyses at the
system scale. However, certain limitations warrant careful attention in the future use of GRAIN dataset:

1.  The performance in identifying secondary and tertiary canals depend widely on the regional coverage of OSM data.
    While certain regions such as San Joaquin Valley, California showed excellent coverage for even tertiary canals, other
    locations such as the Krishna and Godavari delta showed noticeable lack of sufficient OSM data and a corresponding
    dip in performance.
2.  Because many OSM features are digitized from using dry-season imagery, ephemeral canals that carry water for only
    a part of the year may be missed.
3.  Mislabelling of canals as any non-waterway OSM tag such as 'roads', is not tackled in the current workflow.
4.  The temporal lag between OSM edits and GRAIN releases may cause recent canal infrastructural changes to be missed
    until the next release.
5.  GRAIN in its current form does not contain many hydraulic attributes such as canal flow, hierarchy, lining materials,
    or operational status, putting constraints on its direct use for hydraulic modelling without supplementary data.

GRAIN's completeness is ultimately tied to the evolving density and tag quality of community contributions. Despite these
constraints, GRAIN's broader impact is substantial. It is uniquely positioned to enhance the utility of multiple satellite missions
such as SWOT. As SWOT captures surface water extent and elevation globally, GRAIN can contextualize those observations
within irrigation networks, enabling hydrologists and water managers to trace the water delivery chain from dams, reservoirs



or lakes to rivers and finally to the farms via engineered irrigation canals. This synergy opens exciting avenues for monitoring irrigation performance, assessing command area coverage, and improving agricultural water use efficiency under climate stress. GRAIN also opens doors to a wide range of operational and policy applications. Integrated with crop-forecasting platforms, the dataset can help isolate irrigation signals from precipitation-driven yield variability, supporting early-warning

systems for food security. GRAIN data can also be combined with groundwater withdrawal data to identify regions to focus canal expansion and modernisation efforts to alleviate severe groundwater depletion.  At the basin scale, planners can use GRAIN to prioritise canal-lining or modernisation investments by pinpointing reaches subject to high conveyance loss. Finally, because the data is openly licensed, they can inform transboundary negotiations by providing a transparent, shared baseline of surface-water delivery infrastructure.

**5.1. Future Work**

To support long-term community engagement and continuous improvement, the GRAIN codebase is now hosted on a public GitHub repository and will be enhanced with a GitHub Discussions board, an issue tracking system, and community contribution guidelines. These features will enable users to report bugs, request new features, exchange ideas, and contribute improvements directly to the dataset and processing workflow.

A companion GRAIN Visualizer web application is currently being planned to allow users to interactively explore canal geometry, review metadata, and provide feedback. The application will be linked from the GitHub repository to encourage seamless participation and transparency in dataset evolution.

Near-term development for version 1.1., will add few high value metadata, most notably canal hierarchy and network order (upstream to downstream numbering), which are not available in the current release. Looking ahead, GRAIN should be

regenerated by the community on an annual or semi-annual cycle, automatically ingesting the latest OSM dumps and enriching attributes with complementary global products such as SWOT water-surface elevations, crop-type data, and command-area footprints. These systematic enhancement updates carried out by the community can ensure that GRAIN remains a relevant, living, community-driven resource that evolves in lock step with user contributions, advances in Earth-observation data and agricultural water management needs.

**Data and Code Availability**

The python codebase for GRAIN is made available through the GRAIN Git repository at
https://github.com/SarathUW/GRAIN.git (Suresh, S., 2025)

The GRAIN Country Scale data is available for download from Zenodo at https://doi.org/10.5281/zenodo.16786488 (Suresh and Hossain, 2025).



## Author Contributions

SS – conceptualization, data curation, analysis, drafting, review and editing

FH – supervision, fund acquisition, conceptualization, analysis manuscript drafting, review and editing

VM – data curation, manuscript review and editing

NH – data curation, manuscript review and editing

## Competing Interests

The authors declare that they have no competing interests.

### Acknowledgements

The authors thank the NASA Physical Oceanography Program, and the NASA Applied Sciences Water Program for their support.

## Financial Support

This study was supported by a NASA grant 80NSSC24K1644 (Quantifying the net impact of the SWOT mission in improving our Understanding of Human Regulation of Surface Water around the World) of NASA Physical Oceanography program. Additional support from the NASA Applied Science (Earth Science to Action) Water program (80NSSC22K0918) to the corresponding author (Faisal Hossain) is acknowledged





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





# Appendix A

## A.1. Performance Evaluation of the Random Forest Classification and Description of Test-Train Techniques

The training and evaluation of the Random Forest classifier was conducted using the 80-20 train test split and a 10-fold cross validation. In the 80-20 approach, the dataset is randomly divided such that 80% of the samples are used to train the model while the remaining 20% are reserved for testing. In contrast, the 10-fold cross-validation method partitions the dataset into ten equal subsets called folds, using nine folds for training and the remaining one for testing. This process is then repeated ten times, each time with different samples in each subset, reducing variance in performance estimates. This increases robustness
to variance in performance evaluation.

Figure A. 1 showcases the model training performance of the Random Forest classifier. Panel (a) presents the features importance analysis results.  Panels (b) - (d) summarize the classifier's performance using several evaluation metrics. Panel (b) presents accuracy, precision, recall, and F1-score for both validation strategies, macro-averaged across the two classes. Accuracy measures the overall proportion of correctly classified segments, while precision quantifies how many of the
segments predicted as canals were actually canals. Recall, on the other hand, indicates how many of the true canal segments were correctly identified by the model. F1-score is the harmonic mean of precision and recall and reflects the model's ability to minimize both false positives and false negatives.

All metrics exceed 98%, suggesting strong and balanced classification performance. Panel (c) shows the confusion matrix for the 80–20 evaluation, indicating very few misclassifications between rivers and canals. Panel (d) displays the distribution of
accuracy across the 10 folds in the cross-validation test. It can be seen that majority of the folds band around the 0.987 range with a mean accuracy of $0.986 \pm 0.002$, underscoring the model's stability. These results validate the classifier's reliability and the robustness of the selected features, confirming its suitability for large-scale application on OSM waterway datasets in diverse geographic contexts.



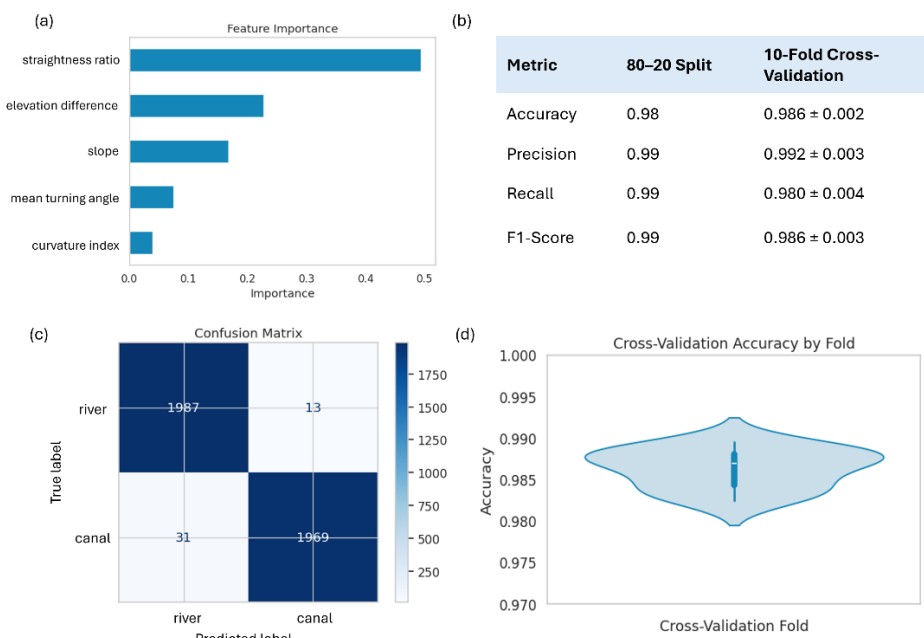


Figure A. 1. Random Forest classification model training performance. (a) Feature importance bar plot showing the relative contribution of each engineered variable to the model, (b) Macro averaged Accuracy, precision, recall, and F1-Scores for 80-20 train-test and 10-fold cross validation test, (c) Confusion matrix for 80-20 train-test, illustrating true and false positives and negatives. (d) Violin plot showcasing the spread of the accuracy scores for the 10-fold cross validation test.






**A.2. Supporting Figures**


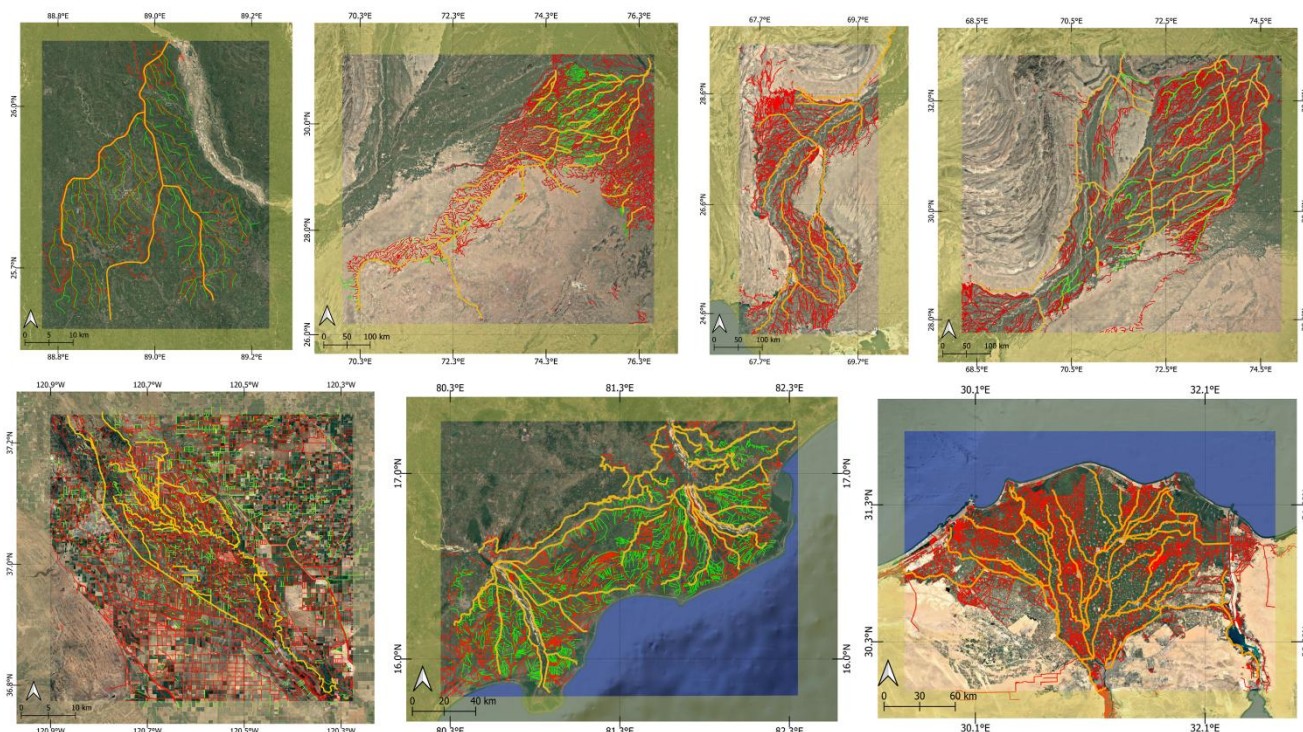

Figure A. 2. In-situ and manually delineated canal network datasets used for the validation of GRAIN dataset. From top left to bottom right - Teesta canal in Bangladesh, Indira Gandhi canal network, Rajathan, India, Lower Indus canal network, Sindh, Pakistan, Upper Indus canal network, Punjab, Pakistan, San Joaquin/Mendota canal network, California, U.S.A, Krishna Godavari canal network, Andhra Pradesh, India,
and the Nile Delta canal network, Egypt. Thicker lines in yellow indicate primary canals, while thinner lines in green indicate secondary and tertiary canals. Lines in red indicate the GRAIN data for the region. This figure complements Figure 6 in the main manuscript, showcasing the validation metrics for the various regions.

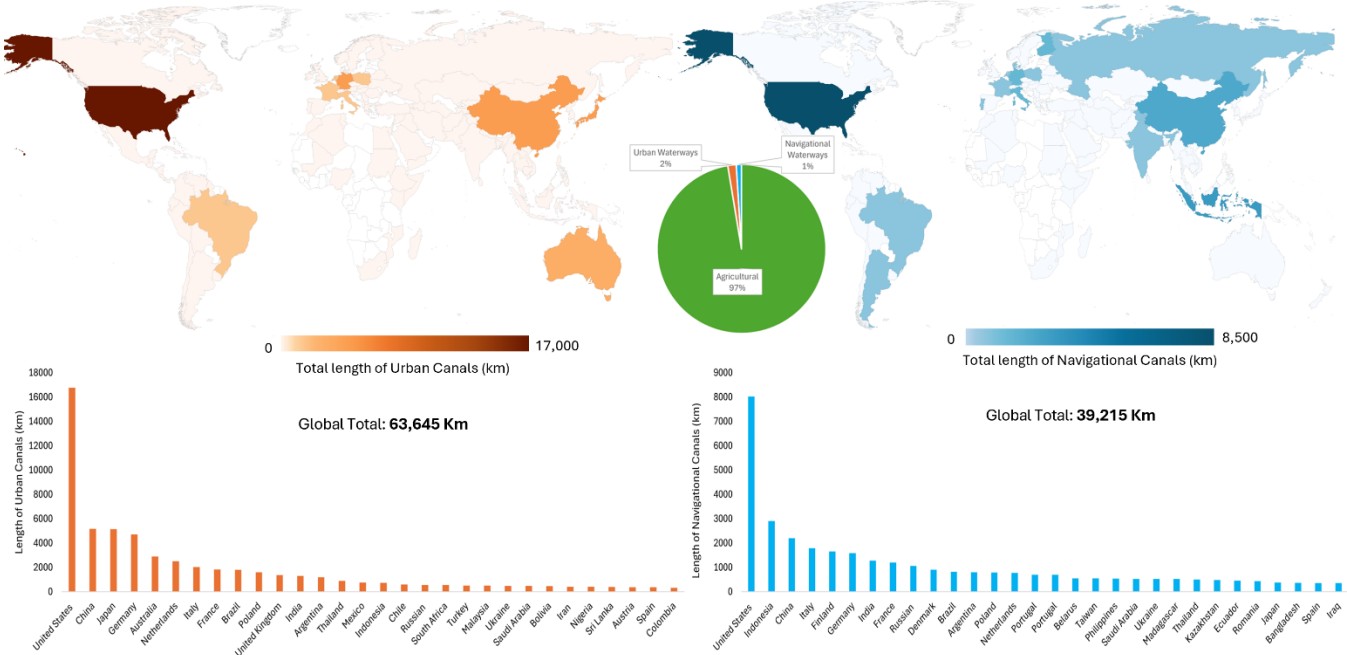

Figure A. 3. Distribution of [left] Urban Canals and [right] Navigational Waterways, with list of top 30 countries. Pie chart in the centre shows the proportion of urban (2%) and navigational canals (1%) to agricultural canals (97%).