# Peer review of "GRAIN - A Global Registry of Agricultural Irrigation Networks"

_Earth System Science Data, 2025_

## Author Comment (AC1)

**Author's Response to Reviewer #1 Comments**

We sincerely thank Reviewer 1 for their thoughtful, constructive, and highly detailed feedback on our manuscript. The overall assessment of the reviewer is that *'the manuscript is technically rigorous and makes a significant contribution to agricultural data, but can be strengthened through some clarifications and revisions.'* We have carefully addressed each comment in an itemized manner in the pages that follow. Reviewer comments are italicized, and our responses are provided in blue colored text. Line numbers referenced correspond to the track changes version of the revised manuscript.

Below, we summarize the major revisions and improvements made to the manuscript in response to the reviewer's comments:

- Improved reproducibility by revamping the GitHub codebase with a sample workflow, launching a documentation website and expanding user guidance for visualizing and usage of GRAIN data.
- Expanded the introduction to more clearly highlight the mapping resolution of GRAIN, and GRAIN's ability to distinguish canal use type.
- Strengthened the feature-engineering section with a clearer discussion of scale-related effects and detailed reporting of polyline-length distributions.
- Enhanced the machine-learning description by adding feature-importance rankings, clarifying hyperparameter tuning, and summarizing model comparisons.
- Tempered the cereal-yield correlation discussion by discussing the influence of other factors and added future work notes on validation of tropical regions.
- Refined cropland buffer logic for canal use classification, clarified the effect of seasonal variation of land cover maps, and explained why precision and F1 score are not a reliable metric for spatial validation.
- Enhanced global figures by adding continent level statistics for canal length and density.

We believe that these revisions, together with the detailed responses that follow, address the reviewer's suggestions comprehensively and substantially improve the overall quality and clarity of the manuscript.

**Detailed Responses to Reviewer #1**

We thank the reviewers for their thorough assessment of our manuscript. The key assessment is that *"The study is technically rigorous, well-structured, and makes a significant contribution to global water and agricultural data infrastructure."* The reviewer, however, recommends a few methodological clarifications and revisions to strengthen the overall paper.

**Reviewer's general assessment**: *'This manuscript presents GRAIN, the first global-scale, open-access vector dataset of irrigation canal networks, developed by integrating OpenStreetMap volunteered geographic information (VGI) with a machine learning classification workflow. The study is technically rigorous, well-structured, and makes a significant contribution to global water and agricultural data infrastructure.*

*The methodological design, particularly the feature engineering and validation framework, is carefully implemented and clearly described. However, a few methodological clarifications would strengthen the paper:*

*The authors should address scale dependency in geometric features (straightness ratio, curvature, mean turning angle), since path segmentation and vertex density can strongly affect these metrics. Validation could include precision and F1-score, in addition to recall, to better characterize model reliability. The interpretation of canal density versus yield should acknowledge potential confounding factors. Some minor details on resampling, path length normalization, and feature computation should be made explicit for full reproducibility.'*

**Our response:** We sincerely thank the reviewer for their thoughtful and positive assessment of our manuscript. We greatly appreciate the recognition of the technical rigor, methodological transparency, and scientific value of the GRAIN dataset. The constructive suggestions provided by the reviewer focus primarily on enhancing methodological clarity, elaborating certain analytical decisions, and strengthening the reproducibility and interpretability of the workflow. We deeply appreciate these and have worked on addressing them, as described in the itemized response below:

**Itemized response to reviewer #1 comments:**
Note that all line numbers mentioned below refer to the track-changes version of the revised manuscript

**Comment 1:** *'Abstract: Consider including the minimum mapping unit or spatial resolution of the GRAIN dataset to provide immediate context for readers.'*

**Our response:** We thank the reviewer for this helpful suggestion. Because GRAIN is a vector dataset derived from OpenStreetMap (OSM), it does not have an inherent spatial resolution in the same way raster datasets do. However, we agree that communicating the effective mapping precision would be valuable context for readers.

To address this, we have added a statement in both the Abstract and the Methods (Section 2.2) clarifying that the effective minimum mapping unit depends on the imagery used by OSM contributors and typically range from ~5 m to 30 m spatial resolution. These changes have been added in between *lines 15-20 and 150-155* in the revised manuscript.

**Comment 2:** *'L90–105: It would be helpful to specify how GRAIN differentiates agricultural irrigation canals from other canal types (urban, navigation, drainage) early on, as this distinction defines the dataset's uniqueness.'*

**Our response:** We thank the reviewer for this suggestion. We agree that introducing the canal-use classification logic earlier in the manuscript will strengthen the reader's understanding of GRAIN's unique contribution.

To address this, we have revised the Introduction section to provide an early description of how canal use types are distinguished. Specifically, in lines 110-115, we now mention that GRAIN integrates OSM waterway vectors with ESA CCI Land Cover data and downstream connectivity checks to differentiate agricultural irrigation canals from urban canals, navigational waterways, and drainage channels.

**Comment 3:** **'***The data sources are clearly summarized. Consider adding a table column for spatial/temporal (if applicable, may include year) resolution (e.g., ESA CCI 300 m, FAO 5' grid, SRTM 30 m) instead of combining type and spatial resolution.'*

**Our response:** We thank the reviewer for this constructive suggestion. We agree that explicitly separating the spatial/temporal resolution information into its own column improves readability and clarity.

In response, we have revised Table 1 to include a dedicated column titled "Resolution", where spatial resolution and temporal information, where applicable, are reported clearly and separately from dataset type.

**Comment 4:** *'L146–153: Clarify whether OSM features were filtered for tagging quality or contributor density, as OSM completeness varies greatly by region.'*

**Our response:** We thank the reviewer for raising this important point. We clarify that no filtering based on OSM tagging quality or contributor density was performed in the current GRAIN

workflow. Information on contributor density, edit history, or tagging reliability is not consistently available across countries in OSM, and we rely on the ML based methodology and other post-processing checks to overcome the tagging quality issues.

We also now explicitly acknowledge this as a limitation of GRAIN in Section 5 with the following added text between *lines 470-475.*

"OSM completeness and tagging quality vary substantially by region, but these metadata are not consistently available and therefore could not be used for filtering. This remains a limitation of GRAIN and may contribute to regional variability in secondary/tertiary canal detection."

**Comments 5, 6, 7, 8, 9:**

**#5 -** '*L225–260: The geometric and topographic feature design is excellent. However, straightness ratio (SR), mean turning angle, and curvature index are inherently scale-dependent — their values can vary with path length or sampling density*'

**#6** - *Please clarify the typical or average polyline length (Dpath) used in training and validation.*

**#7** - *If canal and river polylines vary greatly in length (e.g., 100 m vs 10 km), SR values can be biased since shorter paths naturally appear straighter even in meandering systems.*

**#8** -*Reporting the distribution (range, mean, median) of Dpath would help assess whether SR differences reflect true geometry or feature segmentation.*

**#9** - '*L230–250: Since both mean turning angle and curvature index depend on vertex spacing, please confirm whether all polylines were resampled to a uniform vertex interval before computing these metrics. Suggested insertion: "All polylines were resampled at uniform vertex spacing before computing geometric metrics to ensure that sinuosity differences reflect genuine geometry rather than digitization density or line length.*'

**Our response:** We thank the reviewer for these insightful comments. Because they relate to the same underlying issue, we provide a combined response.

We fully acknowledge that geometric metrics such as straightness ratio, curvature index, and mean turning angle can exhibit scale dependency, particularly when OSM polylines differ in length or vertex density. We also clarify that no vertex resampling was applied, as doing so may distort engineered canal geometries. Any misclassification that may arise from these scale effects is mitigated through a multi-stage post-processing workflow involving (i) removal of segments overlapping known SWORD river centrelines, (ii) correction using OSM name and tag fields, and (iii) iterative connectivity propagation to retain segments that form coherent canal networks.

To address the reviewer's concerns, we have added the following clarifications to Section 2.3.2 (Feature Engineering):

1. Acknowledgment of scale dependency: The following content has been added in lines 260-265 to explicitly acknowledge the scale related variation in some features.
"Geometric metrics such as straightness ratio, mean turning angle, and curvature index may exhibit scale-dependent behavior because OSM polylines vary in length and vertex density. Thus, some variability in these metrics is expected. This misclassification that might be present due to this limitation is addressed by the various post processing checks."

2. Typical DPath values and information on its distribution used in training and validation: We now report the average polyline length for the training and validation datasets, along with their inter quartile range as shown below:

| | Training Data | | Validation Data |
|---|---|---|---|
| | Canals | Rivers | Canals |
| Average polyline length | 1.6 km | 58.8 km | 12.63 km |
| Inter Quartile Range (25% - 75%) | 0.02 km – 2.4 km | 20.7 km – 83.7 km | 2.62 km – 14.1 km |

These values indicate that OSM canals and rivers differ in their typical path lengths. Canals are represented as shorter, segmented engineering reaches (mean 1.6 km), whereas rivers form long, continuous natural drainage paths (mean 58.8 km). This systematic difference explains why geometric metrics such as straightness ratio naturally separate the two classes. Although scale dependency exists, it reflects real-world geometric distinctions rather than any artifacts. The longer path lengths in the validation datasets (mean 12.6 km) arise because many validation networks contain only major primary canals.

These details have been added to the revised manuscript between lines 265-270 for the training data and lines 365-370 for the validation data.

3. Clarification of potential bias from short vs. long polylines:

Regarding potential bias from shorter versus longer polylines, we have clarified in lines 260-270 that although shorter segments tend to appear straighter, it is a natural reflection of the real world properties of canals and rivers. Also, the classifier does not rely on straightness ratio alone. The Random Forest model uses multiple complementary features, including elevation difference and slope, which help counterbalance any scale-related variability in geometric metrics. These features collectively allow the model to

distinguish canals from rivers robustly, even when individual geometric metrics exhibit scale dependency.

**Comment 10**: *'L265–275: Consider moving feature importance rankings (e.g., straightness ratio > slope > elevation) to the main text instead of supplementary dataset. State explicitly whether hyperparameters (number of trees, max depth, etc.) were tuned or used as defaults. It would also be better if the authors had model comparison and selection.'*

**Our response:** We thank the reviewer for this helpful suggestion. In accordance with the suggestions, we have made the following changes:

1. The feature-importance ranking has been presented in the main text as Table 2 in the revised manuscript.

2. Hyperparameters explicitly stated:
   We now clearly specify between lines 280-285 that only the number of trees was tuned (set to n = 200) based on iterative testing for stability and generalization. All other hyperparameters, including maximum depth, minimum samples per split, and number of features considered at each split, used default scikit-learn values. This has been added in Section 2.3.2.

3. Model comparison clarified:
   To address the reviewer's request for model selection context, we added the following statement in the main text in lines 290-295.

   "The Random Forest model was compared to other popular models such as XGBoost and CatBoost to compare model performance. Comparable performance was observed across all three classifiers (overall accuracy > 98%). Thus, the Random Forest was retained for the GRAIN workflow due to its interpretability, computational efficiency, and stability across diverse geographies.

**Comment 11**: *'L295–310: Please clarify: How were mixed land-cover pixels (e.g., cropland + urban) within the 1 km buffer handled (such as cities in Guangzhou and Shanghai)? Could seasonal cropland variability affect classification? If so, consider acknowledging this as a source of uncertainty. When propagating connectivity from cropland-linked canals, specify whether a distance threshold was applied to avoid overextension into non-irrigated zones'*

**Our response:** We thank the reviewer for raising these important points. We clarify that buffers containing both cropland and urban areas are still classified as agricultural as long as cropland remains the dominant class. This is now noted explicitly in Section 2.3.4. in lines 320-325.

We agree that cropland seasonality may affect classification, since ESA CCI LC represents a single-year composite and does not capture intra-annual variability. We now acknowledge this as a source of uncertainty and has been reported in Section 5, lines 475-480.

Finally, to prevent overextension of irrigation classification into non-irrigated zones, canal segments that are not majority-cropland are only promoted to "irrigation canal" status if they are end-to-end connected to a previously identified irrigation canal within a maximum connection tolerance of 100 m. We have added this explicit threshold (100 m) in Section 2.3.4. lines 325-330.

**Comment 12:** *'L325–345: The use of recall and mean offset distance (MOD) is appropriate. However, for a full picture of model reliability, please report precision and F1-score as well.*

**Our response:** We thank the reviewer for this suggestion. While we agree that precision (false positive rate) and F1-score are informative metrics, they cannot be reliably computed for the purpose of validation

The reference canal networks used for validation, whether national inventories or manually delineated maps, might not have the complete collection of true canals. This is particularly true for manually delineated datasets which include only the major canals in the region, whereas GRAIN contains many smaller secondary and tertiary canals. Using these validation datasets to compute precision would therefore treat every detected secondary/tertiary canal as a false positive, leading to a systematic underestimation of precision and F1-score. In addition, the validation datasets were created at different points in time. Thus, in many regions, more recent OSM edits include newly mapped canals that were not present when the validation data were compiled. These legitimately mapped canals would again be erroneously treated as false positives, further biasing precision downward.

For this reason, we strongly believe that recall is the only appropriate and unbiased metric for spatial validation. Precision and F1-score are however, already reported for the model training and cross-validation stages where complete labels are available.

**Comment 13:** *'L340–345: Provide sample sizes (km of validation canals) for each region (e.g., Nile Delta, Indira Gandhi Canal) and note the proportion of primary vs. secondary canals.'*

**Our response:** We have added the relevant information regarding sample sizes and proportion of primary vs secondary canals in Figure 6 of the revised manuscript.

**Comment 14:** *'L345–350: Consider adding at least one tropical validation region (e.g., Southeast Asia) to test robustness under different geomorphological and land-cover contexts'.*

**Our response:** We thank the reviewer for this thoughtful suggestion. We agree that expanding validation to include additional tropical regions would further strengthen the demonstration of GRAIN's robustness across diverse geographic and land-cover settings. The current version of GRAIN includes comprehensive validation over several large and hydrologically complex regions, including the Teesta Basin (Bangladesh), Indira Gandhi Canal (India), Krishna–Godavari Delta (India), the Indus Basin (Pakistan), the Nile Delta (Egypt), and the San Joaquin/Mendota system (USA), covering a wide range of climatic, topographic, and land-use conditions.

However for the southeast Asia regions, high-quality canal reference datasets are not readily available. Given the scope of the current version (v1.0), we have not added a new tropical validation region in this release.

We have now added a statement in the Future Work section in lines 500-505, noting that future versions of GRAIN, particularly through community contributions, may incorporate additional validation across tropical climates and other land-cover categories, including regions where OSM completeness varies significantly.

**Comment 15:** *'L355–385: Figures 7–9 effectively illustrate global canal distribution. Consider adding a continent-level summary table (total canal length and density) to complement global maps.'*

**Our response:** The continent level summary for both total length and density has been added as additional insets in figures 7 and 9 of the revised manuscript.

**Comment 16:** *'The correlation between canal density and cereal yield (r = 0.31) is interesting but weak. Please emphasize that this is indicative, not causal, and may be influenced by other variables (e.g., irrigation efficiency, water management, input use). A partial correlation or multivariate regression would strengthen this analysis.'*

**Our response:** We agree that the reported global correlation between canal density and cereal yield (r = 0.31) is only indicative and should not be interpreted as causal. We have now clarified in the manuscript between lines 410-420 that this association likely reflects the influence of many unobserved factors, including irrigation efficiency, water management practices, crop inputs, technology adoption, and groundwater use, that vary widely across countries and can strongly affect yield independently of canal density.

**Comment 17:** *'L410–430: The discussion nicely links GRAIN to SWOT and remote-sensing applications. Consider adding a brief quantitative uncertainty estimate (e.g., expected completeness by region or OSM coverage density).'*

**Our response:** We thank the reviewer for this thoughtful suggestion. We agree that adding a quantitative perspective on uncertainty would strengthen the Discussion. However, two

methodological constraints make it challenging to provide a quantitative completeness estimate or OSM coverage density at the global scale

1. Lack of global ground-truth canal inventories
   There is no global benchmark dataset of irrigation canals against which completeness could be quantified. Existing national inventories (e.g., NHD, India Canal Network) cover only specific countries, and manual delineations exist only for selected basins.

2. OSM contributor density and tagging completeness are not uniformly available
   OSM does not systematically publish global, spatially explicit measures of contributor activity or tagging quality for canals. Coverage density also varies by region, imagery availability, and community engagement, making global metrics inconsistent and unreliable.

We have strengthened the manuscript by adding a qualitative uncertainty discussion in Section 5, lines 480-485 stating:

"Because the completeness of OSM waterway mapping varies geographically and cannot be quantified uniformly across countries, GRAIN completeness may differ across regions, particularly for secondary and tertiary canals. Regions with dense OSM activity (e.g., Europe, USA) generally show higher capture rates than areas with sparse contributions (e.g., parts of Africa or Southeast Asia)"

**Comment 18**: *'L470–475: To enhance the reproducibility of the dataset generation process, it would be helpful if the authors could reorganize the code repository so that data acquisition is more automated and user-friendly. Specifically, the authors are encouraged to either include the necessary reference data (or sample subsets) within the repository's assets folder, or update the code to automatically request and download required datasets from their original sources. In addition, please update the README file to provide clear, step-by-step guidance for users to reproduce the workflow, from data retrieval to model execution and dataset generation.'*

**Our response:** We thank the reviewer for this helpful suggestion aimed at improving reproducibility. In response, we have revamped the GRAIN GitHub repository to include a new sample workflow that demonstrates how to reproduce the GRAIN dataset for a representative region in Egypt.

Instead of updating the README file of the Git repository, we have created a dedicated documentation website, available at https://grain-canals.readthedocs.io , which provides clear step-by-step instructions for reproducing GRAIN, including data acquisition, preprocessing, feature engineering, model execution, and final output generation. All prerequisite datasets

required for the sample workflow are now hosted in a separate Zenodo repository (https://zenodo.org/records/17608198). In addition, the documentation site now contains detailed information on the GRAIN attribute schema, a high-level description of the methodology, guidance on reading and visualizing GRAIN data, and best practices for working with the dataset. The addition of the documentation website has been reported in the revised manuscript in lines 515-520 in the Data and Code Availability section. These updates are aimed at improving the usability and reproducibility of the GRAIN workflow and directly address the recommendation.

---

## Author Comment (AC2)

**Author's Response to Reviewer #2 Comments**

We sincerely thank Reviewer 2 for their thoughtful and constructive review of our manuscript. The reviewer noted that the GRAIN dataset is valuable and timely, with a logically structured and robust methodology, while identifying several points that would benefit from clarification and contextualization. We have carefully addressed each comment in an itemized manner. Reviewer comments are italicized, and our responses are provided in blue colored text. Line numbers referenced correspond to the track changes version of the revised manuscript.

Below, we summarize the major revisions and clarifications made to the manuscript in response to Reviewer 2's comments:

- Explicitly acknowledged potential regional bias arising from the concentration of training data in United States and India.
- Clarified the sampling strategy underlying feature distributions shown in Figure 4.
- Clarified the rationale for using Mean Offset Distance (MOD) as a validation metric.
- Explained the handling of transboundary canals in country-level statistics.
- Expanded the Future Work section to highlight the value of incorporating canal lining and other canal attributes.

We believe that these revisions and clarifications address Reviewer 2's comments and further strengthen the transparency, rigor, and usability of the GRAIN dataset.

**Detailed Responses to Reviewer #2**

We thank the Reviewer for their thoughtful and constructive assessment of our manuscript. The reviewer notes that "the manuscript is logically structured, clearly presented, and robust in its methodology," and that the resulting GRAIN dataset is "highly valuable and timely, effectively filling a significant gap in global-scale irrigation infrastructure data." The reviewer also raises several specific points aimed at clarifying methodological choices, validation metrics, and the interpretation of selected results.

**Reviewer's general assessment**: *'This paper presents a machine learning-based classification workflow to reclassify OSM hydrographic data, successfully extracting a global network of agricultural irrigation canals. This study and the resulting Global Irrigation Canal (GRAIN) dataset are highly valuable and timely, effectively filling a significant gap in global-scale irrigation infrastructure data. Overall, the manuscript is logically structured, clearly presented, and robust in its methodology. However, I have several specific concerns that need to be addressed'*

**Our response:** We sincerely thank the reviewer for this positive evaluation of the scientific contribution and relevance of the study. We have carefully gone through every point raised by the reviewer and have addressed these through targeted clarifications and revisions to the manuscript, as detailed in the itemized responses below.

**Itemized response to reviewer #2 comments:**
Note that all line numbers mentioned below refer to the track-changes version of the revised manuscript

**Comment 1:** *'The training data comes primarily from canal datasets in the United States and India, which may introduce regional bias and affect the model's generalization ability in other global irrigation systems. This limitation should be explicitly stated in the discussion.'*

**Our Response:** We thank the reviewer for highlighting this important point. We agree that the training data used to develop the Random Forest classifier is primarily derived from national-scale canal inventories in the United States and India, and that this may introduce regional bias that could influence model generalization across diverse global irrigation systems.

We have now explicitly acknowledged this limitation in the Discussion section of the revised manuscript between **lines 500-510**. We clarify that while the canal systems in the United States and India encompass a wide range of canal characteristics, including different geometries, sizes, and management practices, the training data does not fully capture the global diversity of irrigation infrastructure. As a result, some degree of regional bias may be present, particularly in regions where irrigation canals differ substantially in geometry, construction practices, or mapping completeness in OpenStreetMap.

**Comment 2:** '*Figure 4 illustrates significant distinctions between natural rivers and artificial canals across five geometric and topographic features (e.g., sinuosity/straightness ratio, slope, turning angle). Did the authors ensure that the sampled rivers and canals were located within similar geographical or topographical environments during this comparison?*'

**Our Response:** We thank the reviewer for this clarification. We would like to note that strict geographical matching was not done between river and canal geometries used for training. Figure 4 presents feature distributions computed using the full set of 20,000 training samples used to train the Random Forest classifier. These samples were drawn through random sampling from all locations where reliable in-situ reference data were available. For canal samples, this corresponds primarily to national-scale inventories in the United States and India, while river samples were drawn from OSM river segments intersecting the SWORD database across all 95 countries included in the GRAIN workflow.

This sampling was intended to expose the classifier to a wide range of geometries, particularly for natural rivers, which can exhibit strong regional variability. While the canal training data spans a more limited set of geographic contexts, irrigation canals are engineered features constrained by design and operational requirements and therefore tend to exhibit more consistent geometric characteristics across regions. As a result, differences in geographic context are less likely to substantially alter the distinguishing geometric signatures between canals and natural rivers. This has been clarified between **lines 225-235**.

**Comment 3:** '*In the validation section, the authors use Mean Offset Distance (MOD) to evaluate the spatial accuracy of the GRAIN dataset. However, the machine learning model designed in this paper acts as a classifier to distinguish whether an OSM segment belongs to a "canal" or a "river", and does not require any geometric coordinate correction of the original OSM vectors. Therefore, the canal positions in the final dataset will fully follow the inherent spatial biases existing in OSM. Why was MOD chosen as the core validation metric to evaluate the "accuracy" of this classification method?*'

**Our Response:** We thank the reviewer for this important clarification and agree with their assessment. As noted, no geometric corrections are applied to the original OpenStreetMap (OSM) vectors in the GRAIN workflow, and the resulting canal geometries therefore inherit any spatial biases present in the source OSM data.

Mean Offset Distance (MOD) is therefore not intended to evaluate the performance of the Random Forest classifier itself, which functions solely as a semantic classifier distinguishing canals from rivers. Instead, MOD is reported alongside recall as a complementary metric to assess the spatial usability of the final GRAIN product when compared against independent in-situ canal datasets that are assumed to have higher positional accuracy. In this context, MOD provides users

with a quantitative indication of the typical spatial offset that may be expected when using OSM-derived, classified canal geometries for regional or global analyses.

This has now been explicitly clarified in the Validation section of the revised manuscript between **lines 370-380.**

**Comment 4:** *'Some canals are transboundary, such as the famous Karakum Canal. When calculating statistics in Figure 9, how were the lengths or proportions of such canals allocated between nations for these statistics?'*

**Our Response:** We thank the reviewer for this clarification. Country-level statistics presented in Figure 9 were computed by aggregating canal lengths within national administrative boundaries. Accordingly, in the case of transboundary canals, such as the Karakum Canal, only the portion of the canal geometry that lies within a given country's boundary contributes to that country's total canal length and associated density statistics.

This boundary-based accounting avoids double counting and ensures consistency in national-level comparisons. We have clarified this approach in the revised manuscript between **lines 455-465.**

**Comment 5:** '*Figure 10 is very interesting, highlighting a non-linear relationship between crop yield and the density of the agricultural canal network.'*

**Our Response:** We thank the reviewer for this positive observation. We agree that Figure 10 does highlight an interesting relationship between cereal yield and canal density. However, we would like to note that the figure is intended to present a high-level contextualization of the association between national-scale agricultural canal density and cereal yield. In line with other reviewer comments, we have also added clarification in the revised manuscript between **lines 430-440** to indicate that the observed relationship is likely influenced by multiple confounding factors and should be treated as indicative, rather than causal.

**Comment 6:** *'If the lining status of the canals could be provided based on the canal network while presenting the global canal network, it would have greater significance for calculating conveyance efficiency in agricultural irrigation and water resource management applications. Of course, this issue is not one that this article needs to solve, but rather a possible direction for the future.'*

**Our Response:** We thank the reviewer for this valuable suggestion and agree that information on canal lining status would substantially enhance the applicability of a global canal network for conveyance efficiency and agricultural water management analyses. At present, canal lining information is not available at a global level in OSM or other open datasets. We have therefore noted this in the future works section between **lines 540-545**.